# A cerebellar internal model calibrates a feedback controller involved in sensorimotor control

Daniil A. Markov [1], Luigi Petrucco [1,2], Andreas M. Kist [1,3] & Ruben Portugues [1,2,4 ✉]

Animals must adapt their behavior to survive in a changing environment. Behavioral adaptations can be evoked by two mechanisms: feedback control and internal-model-based control. Feedback controllers can maintain the sensory state of the animal at a desired level under different environmental conditions. In contrast, internal models learn the relationship between the motor output and its sensory consequences and can be used to recalibrate behaviors. Here, we present multiple unpredictable perturbations in visual feedback to larval zebrafish performing the optomotor response and show that they react to these perturbations through a feedback control mechanism. In contrast, if a perturbation is long-lasting, fish adapt their behavior by updating a cerebellum-dependent internal model. We use modelling and functional imaging to show that the neuronal requirements for these mechanisms are met in the larval zebrafish brain. Our results illustrate the role of the cerebellum in encoding internal models and how these can calibrate neuronal circuits involved in reactive behaviors depending on the interactions between animal and environment.

[1] Sensorimotor Control Research Group, Max Planck Institute of Neurobiology, 82152 Martinsried, Germany. [2] Institute of Neuroscience, Technical University of Munich, 80802 Munich, Germany. [3] Division of Phoniatrics and Pediatric Audiology, Department of Otorhinolaryngology, Head and Neck Surgery, University Hospital Erlangen, Friedrich-Alexander-University Erlangen-Nürnberg, 91054 Erlangen, Germany. [4] Munich Cluster for Systems Neurology (SyNergy), Munich, Germany. ✉email: ruben.portugues@tum.de

The interaction between animals and their surroundings changes constantly, due to changes in the environment and due to processes such as development, growth or injury, which modify the body of the animal. Nevertheless, fine motor control is so important that evolution has provided animals with mechanisms to produce precise behavior in these changing conditions. The task of adapting behavior to the changing environment can be solved in two ways. One way is to react to these changes through a feedback control mechanism, which ensures that the goal of a behavioral act is achieved under a variety of conditions. A second option is for the animal to learn the new environmental conditions, namely the association between its behavior and the sensory feedback, and to adjust its behavioral program in the long-term. This second mechanism is only possible if the change in conditions lasts and can therefore be predicted.

Many stimulus-driven behaviors result in the effective cancellation of the stimulus that evoked them. Examples include the optokinetic reflex (OKR)[1], in which retinal slip evokes eye motion that sets this slip to zero. If the stimulus is monitored constantly, a feedback control loop with well-tuned parameters may provide an appropriate mechanism for performing the task of setting the stimulus to zero[2]. This happens online, so feedback controllers are limited by the time delay required for sensory processing, which in the case of visual feedback is estimated to be between 100 and 300 ms[3–7]. If the processing of sensory information is long with respect to the duration of the motor action, the current state of the body will change dramatically by the time the feedback signal starts to influence the motor command. As a result, the feedback signal will implement an inappropriate correction based on out-of-date sensory information[8].

To overcome this limitation of feedback motor control, the brain can encode internal models of different parts of the body and/or of different aspects of the external world[9–11]. These models monitor the sensorimotor transformation performed by the body during movement and learn a forward or an inverse transfer function of this transformation either to predict sensory consequences of a motor command (forward models), or to provide an appropriate feedforward command to reach a desired sensory state (inverse models). Such models can predict that, for example, if we lift an arm a certain distance, we will experience a certain proprioceptive feedback of this motion (forward model), or that we need to send a certain motor command in order to lift an arm a certain distance (inverse model). If the transfer function changes in a long-term manner, the model updates, leading to motor learning. It is widely believed that internal models for motor control exist in the central nervous system and that in vertebrates, the cerebellum plays a major role in encoding them[8,12–16], although how exactly this is done is not well understood.

In this study, we investigate the interplay between feedback controllers and internal models and the role of the cerebellum in encoding them. We make use of the larval zebrafish optomotor response (OMR)[17], a behavior shared by many animals[18,19], by which they turn and move in the direction of perceived whole-field visual motion. The OMR can be defined in terms of a feedback control mechanism as a locomotor behavior that tries to set the optic flow to zero, thus stabilizing the animal with respect to its visual environment; in this framework, the OMR is similar to the OKR as both of these behaviors effectively cancel the stimulus that evoked them.

As larvae, zebrafish swim in bouts that comprise several full tail oscillations and last around 350 ms, separated by quiescent periods called interbouts[20]. When zebrafish, or any other animal, move forward, they experience the visual scene coming towards them. Previous work has shown that larval zebrafish swimming in a closed-loop experimental assay react to perturbations in this visual feedback[21–23]. Specifically, if a larva receives less feedback than normally, it swims for longer, as if trying to compensate for this lack of feedback by increasing its bout duration. This reaction happens on the time scale of individual bouts[21], so we call this phenomenon "acute reaction". A hypothesized mechanism of acute reaction is that fish use an internal representation of expected sensory feedback, and if the actual feedback does not meet this expectation, they adapt their behavior to minimize this discrepancy[21]. This postulates that fish use forward internal models to compute predicted sensory feedback from motor commands during acute reaction. In a subsequent study, it was further proposed that these predictive computations occur in the cerebellum[22].

Here, we employ behavioral tests and modeling to demonstrate that acute reaction to unexpected perturbations can be implemented by a simple feedback controller, without internal models. The state of this feedback controller can be adjusted if the animal experiences a long-lasting, and therefore predictable, perturbation in sensory feedback. Crucially, loss-of-function experiments have shown that an intact cerebellum was necessary for this recalibration but not for the functioning of the feedback controller itself. We used functional imaging in animals performing adaptive optomotor locomotion to determine whether neuronal requirements of this hypothesis are met in the larval zebrafish brain. Our results illustrate the role of the cerebellum in encoding internal models, which can calibrate existing neuronal circuits according to predictable features of the environment.

## Results

**Unexpected perturbations in visual reafference result in acute behavioral reaction.** When an animal, such as larval zebrafish, moves in a given direction, it naturally experiences optic flow in the opposite direction (Fig. 1a). We will refer to the swimming-elicited velocity of the optic flow as visual reafference (Fig. 1b). To investigate how perturbations in visual reafference affect ongoing behavior, we took advantage of the previously developed closed-loop experimental assay[21,22] (Fig. 1c and Supplementary Movie 1). In this assay, head-restrained zebrafish larvae swim in response to forward whole-field visual flow, such as a forward moving grating, a behavior known as the OMR. A high-speed camera captures this behavior, and the fictive larvae velocity is inferred from the tail motion (see "Closed-loop experimental assay in head-restrained zebrafish larvae" section in Methods). To provide fish with visual reafference, this estimated velocity can be subtracted from the initial stimulus velocity such that the larvae experience the sensory consequences of their own swimming (Fig. 1c, d and Supplementary Movie 1). Importantly, the transformation that determines how estimated swimming velocity translates to reafference is under experimental control, which allows us to introduce perturbations in reafference and to study how the animals react to these perturbations.

In the first set of experiments, we aimed to characterize the acute reactions of zebrafish larvae to a variety of different perturbations and to determine whether these reactions can result from a feedback control mechanism. We used three distinct perturbations in reafference to probe the space of possible behavioral reactions (Fig. 2a and Supplementary Movie 1). The first reafference condition, which has been previously used[21,22], we call *gain change*. It corresponds to changing the gain of the experimental closed-loop, such that larvae receive more or less visual reafference upon swimming (Fig. 2a$_i$). Note that a gain of 0 corresponds to open-loop and a gain of 1 to the freely-swimming condition, referred hereafter as the normal reafference condition (respectively, red and blue shaded bars in Fig. 2a). The *gain change* tests how behavior depends on the amount of reafference that larvae receive upon swimming (e.g., covered distance or reached velocity). The second condition we call *lag*, and this corresponds to introducing an artificial temporal delay between the behavior of the larva and the reafference it experiences

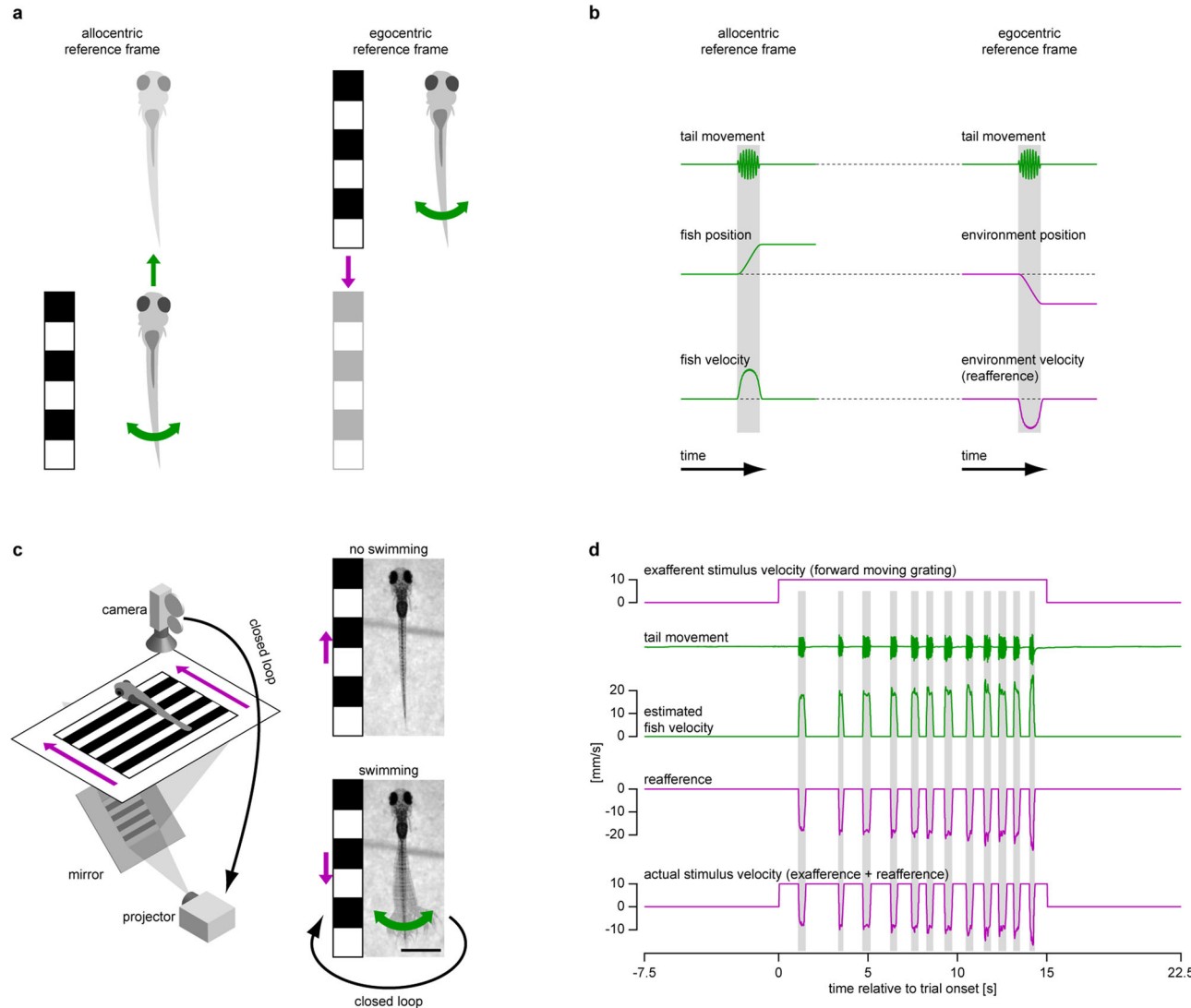

**Fig. 1 Closed-loop experimental assay to study optomotor behavior in larval zebrafish. a** When a larval zebrafish swims forward with respect to its visual environment (left), the environment moves backwards with respect to the fish (right). Variables expressed in motor coordinates, such as tail movement and resulting position or velocity of the swimming larva, are presented in green. Variables expressed in sensory (visual) coordinates, such as observed position or velocity of the visual environment, are presented in magenta. This color-code is used throughout the figures. **b** Change in position and velocity of a swimming fish with respect to its visual environment (left) and of the environment with respect to the swimming fish (right). In all figures, decrease of environment position and velocity along the *y*-axis means that fish progresses forward with respect to the environment. Swimming-elicited change in environmental velocity is referred hereafter as visual reafference, in contrast with externally-generated changes in velocity, referred as exafference. **c** Behavioral rig (left) and schematics of the closed-loop experimental assay (right) used to induce OMR and to provide visual reafference to the fish. Scale bar: 1 mm. **d** Raw data recorded during one experimental trial. **b**, **d** vertical shaded bars indicate swimming bouts.

(Fig. 2a$_{ii}$, top). In the *shunted lag* version of this condition, the reafference is set to zero when the larvae stop swimming (Fig. 2a$_{ii}$, bottom). The *lag* tests how behavior depends on the temporal relationship between the bout and its related reafference. The final reafference condition, *gain drop*, corresponds to dividing the first 300 ms of a bout into four 75 ms segments and setting the reafference to zero during one or more of these segments (Fig. 2a$_{iii}$). The *gain drop* tests whether perturbations in reafference lead to the same behavioral reactions regardless of when they occur within the bout.

Individual wild-type larvae were exposed to 15 s trials during which a grating moved in a caudal to rostral direction at 10 mm/s (Fig. 1d). Larvae responded by performing swimming bouts and the reafference conditions were randomized on a bout-by-bout basis. Perturbation-induced changes in bout and subsequent interbout duration are presented in Fig. 2c, d (black data points).

For all types of reafference conditions, the bout duration increased when the overall reafference was less than normal (Fig. 2c). This was particularly noticeable for the very low gains 0 and 0.33 (Fig. 2c$_i$), under the *lag* and *shunted lag* conditions (Fig. 2c$_{ii}$–c$_{iii}$) and under the *gain drop* conditions where more than one bout segment had gain 0 (Fig. 2c$_{iv}$). Interestingly, the observed increase in bout duration was close to linear as a function of the lag and, as expected, it did not show a significant difference between the *lag* and *shunted lag* cases, as these two conditions were identical during the bout. Finally, under the *gain drop* condition, the mean bout duration was differentially prolonged depending on what bout segment had a perturbed reafference. Overall, a segment with a gain of 0 had a larger effect on increasing the bout duration the earlier it occurred within the bout: compare for example the cases for gain profiles 0111 and 1110 (gray triangles in Fig. 2c$_{iv}$).

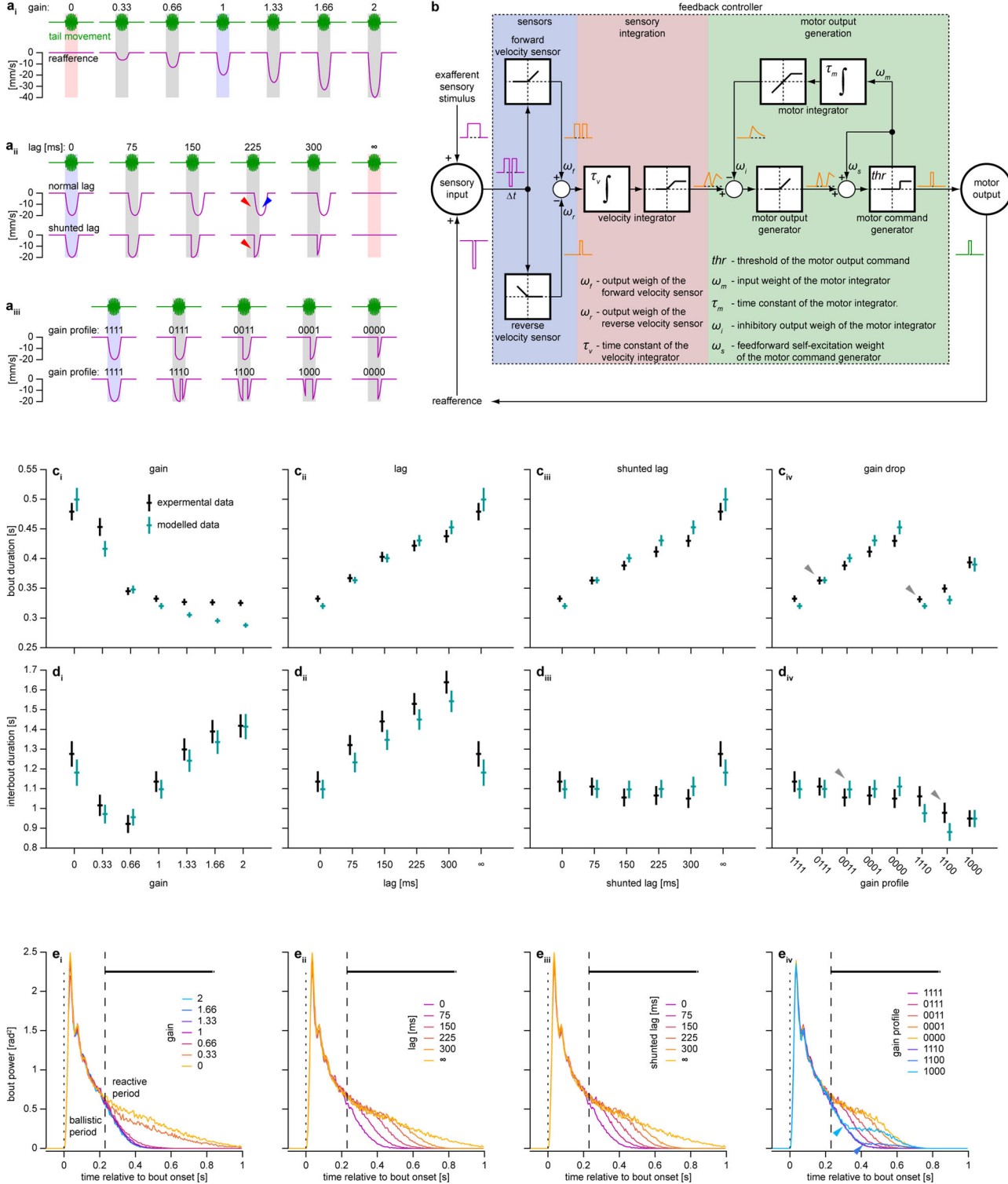

The effects on the interbout duration are displayed in Fig. 2d. Decreasing the gain initially resulted in shorter interbouts (gain 0.66–2) although further decreases reversed this tendency to the extent that interbouts at gain 0 were longer than those at gain 1 (Fig. 2d$_i$). Under the *lag* conditions, the mean interbout duration increased with longer lag in the *non-shunted* setting only (Fig. 2d$_{ii}$–d$_{iii}$). This demonstrates that the duration of a bout and a subsequent interbout can be independently influenced by different aspects of the reafference. Explicitly, insufficient reafference in the beginning of the bout, present in both *lag*

settings (red triangles in Fig. 2a$_{ii}$), increases the bout duration (Fig. 2c$_{ii}$–c$_{iii}$), whereas excessive reafference after the bout end, only present in a *non-shunted lag* setting (blue triangle in Fig. 2a$_{ii}$), lengthens the interbouts (Fig. 2d$_{ii}$). Under the *gain drop* conditions, the interbout duration decreased if the preceding bout had a gain drop (Fig. 2d$_{iv}$). In contrast with the results for mean bout duration, interbouts were affected more when the gain was dropped in segments closer to the end of the bout: compare for example the cases for gain profiles 0011 and 1100 (gray triangles in Fig. 2d$_{iv}$).

**Fig. 2 Acute reaction to unexpected perturbations in visual feedback can be implemented by a feedback controller. a** Reaference conditions used to induce acute reaction: *gains* (**i**), *lags* (**ii**), and *gain drops* (**iii**). Vertical shaded bars indicate swimming bouts, blue and red bars indicate normal reaference and open-loop conditions, respectively. Red triangles indicate insufficient reaference in the beginning of the bout, blue triangle indicates excessive reaference after the bout offset. *Gain drop* conditions (**a**$_{iii}$) are labeled by four digits indicating the gain during the corresponding bout segments (e.g., condition 1100 has normal reaference during the first 150 ms of the bout but no reaference for the next 150 ms). **b** Feedback control model of acute reaction. White squares depict mathematical operations performed by respective nodes: integration, rectification, saturation, and thresholding. Magenta and green traces represent input and output of the model, orange traces represent output of respective nodes. Seven small Greek letters and *thr* denote eight parameters of the model; $\Delta t$ denotes sensory processing delay of 220 ms. **c, d** Mean bout duration (**c**) and interbout duration (**d**) as a function of *gain* (**i**), *lag* (**ii**), *shunted lag* (**iii**) and *gain drop* profile (**iv**) for larvae (black) and the model (cyan). To obtain data for one larva, bout and interbout durations were averaged within each reaference condition. Mean ± SEM across larvae/models is shown; $N = 100$. Gray triangles in **c**$_{iv}$ and **d**$_{iv}$ indicate two gain drop conditions, in which the gain was set to 0 during the same number of 75 ms segments of a bout but the behavior was modified differently depending on what bout segment had a perturbed reaference. **e** Bout power profile as a function of *gain* (**i**), *lag* (**ii**), *shunted lag* (**iii**) and *gain drop* profile (**iv**), averaged within each reaference condition in each larva. Median across larvae is shown. Dotted lines indicate bout onsets, dashed lines separate ballistic and reactive periods. Thick horizontal black lines above the plots indicate time points, at which bout power depends on reaference condition (Kruskal–Wallis test, $p < 0.05/220$, where 220 is the total number of tested time points). Blue triangles indicate that if perturbation in reaference was introduced after the bout had already started (as in gain drop conditions 1000 and 1100), the deviation in the respective mean bout power starts only around 220 ms after the start of the perturbation.

---

In summary, these results demonstrate that larval zebrafish acutely react to unexpected perturbations in visual reaference in an intricate way. Bout duration is prolonged if the reaference is insufficient or delayed compared to the normal reaference condition, and reaference during the early segments of swimming bouts has more influence over the bout duration. On the other hand, larvae prolong the interbout duration if they receive excessive reaference during or after the preceding bout, with reaference during late bout segments having more influence.

**Acute reaction is implemented after a sensory processing delay.** We hypothesized that this acute reaction is implemented by a feedback controller, which must rely on a relatively slow measurement of the controlled variable (see "Introduction" section). Therefore, this acute reaction should be implemented only after a sensory processing delay. To identify the time that larval zebrafish need to react to unexpected perturbations in reaference, we analyzed the temporal dynamics of the tail beat amplitude in the form of bout power within individual bouts in different reaference conditions (Fig. 2e; see "Behavioral data analysis" section for details).

Comparing the mean bout power profiles across different reaference conditions revealed that, when the reaference was perturbed from the very beginning of a bout (as in *gain change*, *lag* and *shunted lag* conditions), larvae reacted by increasing the tail-beat amplitude only 220 ms after the bout onset (Fig. 2e$_i$–e$_{iii}$). However, if the change in the reaference was introduced once the bout had already started (as in the *gain drop* condition with gain profiles 1000, 1100), the deviation in the respective mean bout power was observed only around 220 ms after the start of the perturbation (blue triangles in Fig. 2e$_{iv}$).

We conclude that larval zebrafish react to perturbations in visual reaference with a delay of 220 ms. This result prompted us to define two periods within bouts: an initial stereotyped ballistic period lasting 220 ms and a subsequent reactive period. An unexpected change in reaference condition (regardless of whether the change occurs during the ballistic or the reactive period), can only affect the tail-beat amplitude during the reactive period (Fig. 2e). Such a prominent sensory processing delay suggests that the OMR is implemented by a feedback controller. Finally, we note that the delay duration is consistent with processing time of visual feedback information in other species[3–7].

**Acute reaction can be implemented by a feedback controller that integrates the optic flow.** To confirm that acute reaction is implemented by a feedback controller, we took a simulation approach, which involved designing such a controller and testing its performance under the aforementioned perturbations in reaference.

The main rationale of the designed model derives from the definition of the OMR in terms a feedback control mechanism, as a locomotor behavior that tries to keep perceived optic flow at zero. If optic flow is constant, an animal moving in discrete bouts cannot achieve this goal at all possible points in time. Instead, it can stabilize its position on average by integrating the optic flow in time, estimating displacement with respect to the visual environment over a time window and performing bouts whenever the integrated signal reaches a threshold.

Following this reasoning, we designed a feedback controller consisting of three parts: a sensory part, a sensory integration part and a motor output generation part (Fig. 2b). The sensory part instantaneously combines forward and backward grating velocity with independent excitatory and inhibitory weights, respectively. This weighted sensory input is then integrated in time by a velocity integrator. The output of this integrator can be interpreted as a metric of motivation to swim, which we refer to as sensory drive. This sensory drive is then fed into a motor output generator that produces a motor command when it reaches a threshold. As zebrafish larvae swim in discrete bouts, the model contains a motor integrator that integrates the motor command in time and inhibits the motor output generator, which eventually leads to the termination of the bout. The output of the motor integrator can be interpreted as a metric of "tiredness" that encapsulates possible peripheral or central mechanisms for bout termination, such as muscle fatigue or modulation of the locomotor central pattern generator. Effectively, the motor integrator ensures that bouts have finite length even when the sensory drive to continue swimming is very high. Finally, to ensure that bouts last for some minimum time once started in cases when the sensory drive becomes low immediately after bout onset (for example, if the gain of the closed-loop is very high and the fish receives a lot of reaference), we introduced a self-excitation loop to the motor output command generator (see "Feedback control model of acute reaction" section in Methods for further details; see also Supplementary Fig. 1a for formal mathematical description of the model).

We fit the model with a set of parameters such that it generated bouts and interbouts of realistic duration in response to forward moving grating in the normal reaference condition (Supplementary Fig. 1a; for distributions of obtained parameter solutions see Supplementary Fig. 1b).

Furthermore, the behavior of the model under different perturbations in reaference reproduced the findings presented

in Fig. 2c, d, including the increased motor output in response to decreased or delayed reafference, the difference in reaction of interbout duration to shunted and non-shunted lags, and different reactions under the *gain drop* condition depending on which bout segment had a perturbed reafference (Fig. 2b–d, cyan data points) (see "Discussion" for details on how these subtleties of acute reaction can be explained by the model). Therefore, acute reaction to perturbed reafference can be implemented by a feedback control mechanism that relies on temporal integration of the optic flow.

**Larval zebrafish are able to integrate the optic flow.** To test the main assumption of this model, namely, the existence of the temporal integration of the optic flow in the larval zebrafish brain, and to gain some further insight into its biological implementation, we performed whole-brain functional imaging in head-restrained larvae expressing GCaMP6s in all neurons[24], while they were performing the OMR in a custom-built light-sheet microscope (Fig. 3a and Supplementary Movies 2 and 3).

After segmenting the imaged brains into regions of interest (Fig. 3b) (ROIs, $N = 24{,}677 \pm 4{,}811$, mean ± SEM across six imaged larvae) (see "Functional imaging data analysis" section in Methods for details), we observed ROIs that increased their fluorescence at the onset of the moving grating (sensory ROIs; see gray triangle in Fig. 3c) or when the larvae were performing bouts (motor ROIs; see black triangle in Fig. 3c; see also Supplementary Movie 2). Analysis of the mean grating-triggered or bout-triggered fluorescence (Fig. 3d, e) revealed that the activity of a vast majority of detected ROIs was either sensory-related, or motor-related (respectively, $48 \pm 5\%$ and $50 \pm 4\%$, mean ± SEM across imaged larvae, $N = 6$). Motor ROIs were located predominantly in the hindbrain and in the nucleus of the medial longitudinal fascicle in the midbrain, and sensory ROIs were mostly present in the hindbrain, midbrain, and diencephalic regions, including the inferior olive, dorsal raphe and surrounding reticular formation, optic tectum, pretectum, and thalamus (Fig. 3f, see also Supplementary Fig. 2 for anatomical reference).

As one of the main assumptions of the model is the existence of optic flow-integrating ROIs, we determined whether some of the identified sensory ROIs display properties of sensory integrators (Fig. 3g–i). To this end, we fitted a leaky integrator model to the fluorescence trace of each sensory ROI and estimated their integration time constants (see "Functional imaging data analysis" section for details). We found that the time constants followed a continuous distribution and varied greatly across sensory ROIs (mean = $2.1 \pm 0.3$ s; SD = $0.9 \pm 0.1$ s; $Q_1 = 1.6 \pm 0.2$ s; $Q_2$ (median) = $2.1 \pm 0.2$ s; $Q_3 = 2.5 \pm 0.4$ s; mean ± SEM across larvae; see inset in Fig. 3i). Fluorescence traces of sensory ROIs with short time constants increased faster after the grating onset and decayed faster when the grating stopped moving compared to ROIs with longer time constants (see Fig. 3g, h for activity of two example sensory ROIs in one trial and averaged across trials, respectively, and see Fig. 3i for trial-average activity of all sensory ROIs). ROIs with short and long time constants were located in distinct brain regions. ROIs with short time constants were located predominantly in the optic tectum and in the inferior olive, whereas ROIs with longer time constants occupied the dorsal raphe with surrounding reticular formation and aforementioned diencephalic regions (Fig. 3j, see also Supplementary Fig. 2 for anatomical reference). This spatial distribution was significantly consistent across imaged larvae (Supplementary Fig. 2, see "Functional imaging data analysis" section for details).

We conclude that certain regions of the larval zebrafish brain (dorsal raphe, pretectum, and thalamus) integrate the velocity of the moving grating in time, and could therefore compute the sensory drive that evokes the OMR (as shown also for other experimental paradigms by refs. [25,26]). This provides an important substrate for the feedback controller-based mechanism of acute reaction presented in Fig. 2.

**Larval zebrafish adapt their behavior in response to a long-lasting perturbation in visual reafference.** We next hypothesized that the role of the cerebellum is to update the state of the neuronal controller of the OMR if the relationship between the behavior and the resulting sensory consequences changes in a consistent and predictable manner, and can therefore be captured in an internal model. To test this hypothesis, we first developed a long-term adaptation experimental assay, in which zebrafish larvae performed the OMR and experienced a long-lasting and consistent perturbation in visual reafference. The paradigm consisted of 240 trials that were grouped into four phases: calibration, pre-adaptation, adaptation, and post-adaptation (Fig. 4a; see "Experimental protocols" section for details). Animals were randomly divided into two experimental groups: normal-reafference control and lag-trained. Lag-trained animals received a constantly lagged reafference during the adaptation phase (225 ms non-shunted lag; Fig. 4a, red trace). Importantly, this assay allows to monitor both acute reaction to unexpected perturbation in reafference (immediately after the perturbation is presented to a naïve larva) and potential long-term behavioral changes (after the larvae experiences the perturbation for some time).

We analyzed the duration of first bouts in each trial, as the first bout should not depend on the putative short-term sensorimotor memory accumulated during the current trial, and should therefore reflect potential long-term changes in the OMR circuitry more clearly than subsequent bouts. As expected, naïve lag-trained larvae acutely reacted to unexpected lag in reafference by increasing their bout duration in the beginning of the adaptation phase (dark-blue arrows in Fig. 4b–d). However, by the end of the adaptation phase, the magnitude of the acute reaction had decreased substantially so that their bout duration became less distinguishable from the normal-reafference control group that was never exposed to perturbed reafference (cyan arrows in Fig. 4b, c, e). This indicates that the circuitry underlying the OMR and its acute reaction to perturbed reafference was recalibrated during long-term exposure to a novel reafference condition. This interpretation is confirmed by a prominent decrease in the bout duration of lag-trained animals observed during the post-adaptation phase, referred to as the "after-effect" (orange arrows in Fig. 4b, c, f).

When we analyzed the modulation of the tail-beat amplitude within individual bouts during the experiment, we confirmed our previous observations presented in Fig. 2e. Thus, acute reaction to lagged reafference in the beginning of the adaptation phase occurred only after a considerable sensory processing delay (Fig. 4g; compare black and dark-blue traces in Fig. 4g$_{ii}$, acute reaction is indicated by magenta arrows). On the other hand, tail-beat amplitude during the initial ballistic period was not modulated during acute reaction (Fig. 4g, h). Since the tail-beat amplitude during the ballistic period does not depend on the current reafference, it can be used as a readout of the homeostatic state of the neuronal controller that determines how forward motion of the grating is transformed into optomotor behavior. If, according to the proposed hypothesis, this state updates during the long-term adaptation, one could expect that the ballistic power would change during the long-term exposure to a novel reafference condition. As predicted by the hypothesis, we observed that the ballistic bout power during the post-adaptation phase was indeed significantly larger than during the

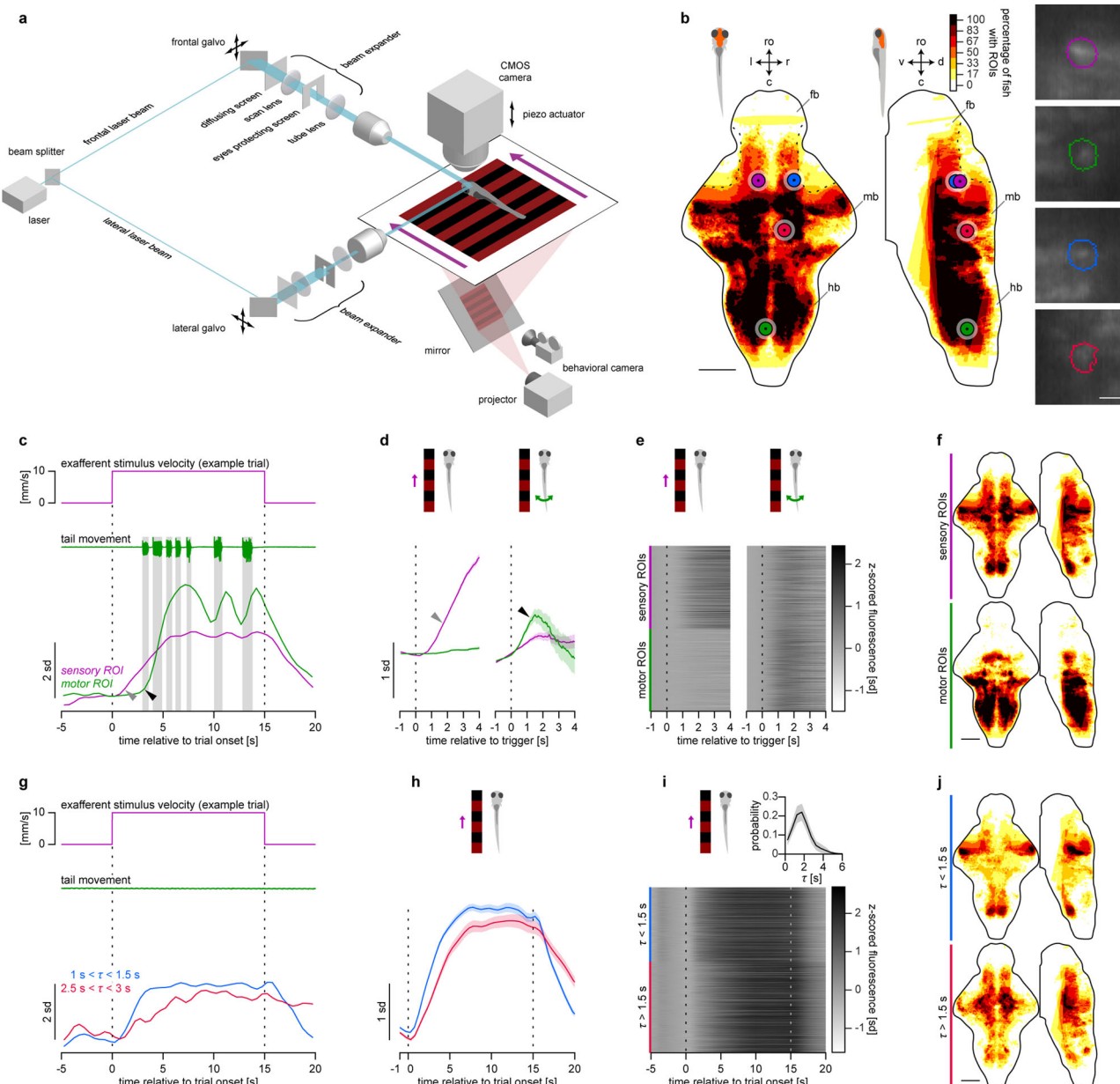

**Fig. 3 Larval zebrafish are able to integrate the optic flow. a** Light-sheet microscope combined with a behavioral rig used in the whole-brain functional imaging experiments. **b** Spatial distribution of detected ROIs from six larvae. Color indicates percentage of larvae with ROIs in the corresponding voxel of the reference brain. **b**, **f**, **j** Maximum dorsoventral or lateral projections are shown: ro rostral direction, l left, r right, c caudal, d dorsal, v ventral; fb forebrain, mb midbrain, hb hindbrain (see Supplementary Fig. 2 for anatomical reference); scale bars: 100 μm. Dotted black curves outline the dorsal rostrolateral midbrain that was blocked from the scanning laser by the eye-protecting screens. Colored circles indicate location of example ROIs shown in the figure, images on the right show their shape (scale bar: 10 μm). **c** Z-scored fluorescence of sensory and motor example ROIs in one trial. **b**–**f** Magenta and green colors represent sensory and motor ROIs, respectively. Vertical shaded bars indicate bouts. Gray triangle indicates the response of the sensory ROI to the grating motion, black triangle indicates the response of the motor ROI after the first bout onset. **d** Average grating-triggered and bout-triggered fluorescence of example ROIs presented in **c**. **d**, **h** Shaded areas represent SEM across triggers. **e** Average grating-triggered and bout-triggered fluorescence of all sensory and motor ROIs pooled from all imaged larvae. **f** Spatial distribution of sensory and motor ROIs. **f**, **j** color indicates percentage of larvae with ROIs from respective functional group in the corresponding voxel of the reference brain, color map is the same as in **b**. **g** Z-scored fluorescence of two sensory ROIs in one trial: one with short integration time constant and one with long. **b**, **g**–**j** Blue and red colors represent sensory ROIs with short and long time constants, respectively. **h** Average grating-triggered fluorescence of example ROIs presented in **g**. **i** Average grating-triggered fluorescence of all sensory ROIs. Inset shows distribution of time constants of sensory ROIs (mean ± SEM across larvae). **j** Spatial distribution of all sensory ROIs with short and long time constants.

pre-adaptation phase (Fig. 4g, compare black and orange traces, increase in ballistic power is indicated by black arrows). Interestingly, this metric of long-term adaptation did not depend on the reafference condition presented during the adaptation phase (Fig. 4j; see "Discussion" section).

These results demonstrate that larval zebrafish are able not only to acutely react to unexpected perturbation in reafference but also to adapt their behavior in the long-term if this perturbation is persistent. The long-term adaptation includes a decrease in magnitude of the acute reaction during the adaptation

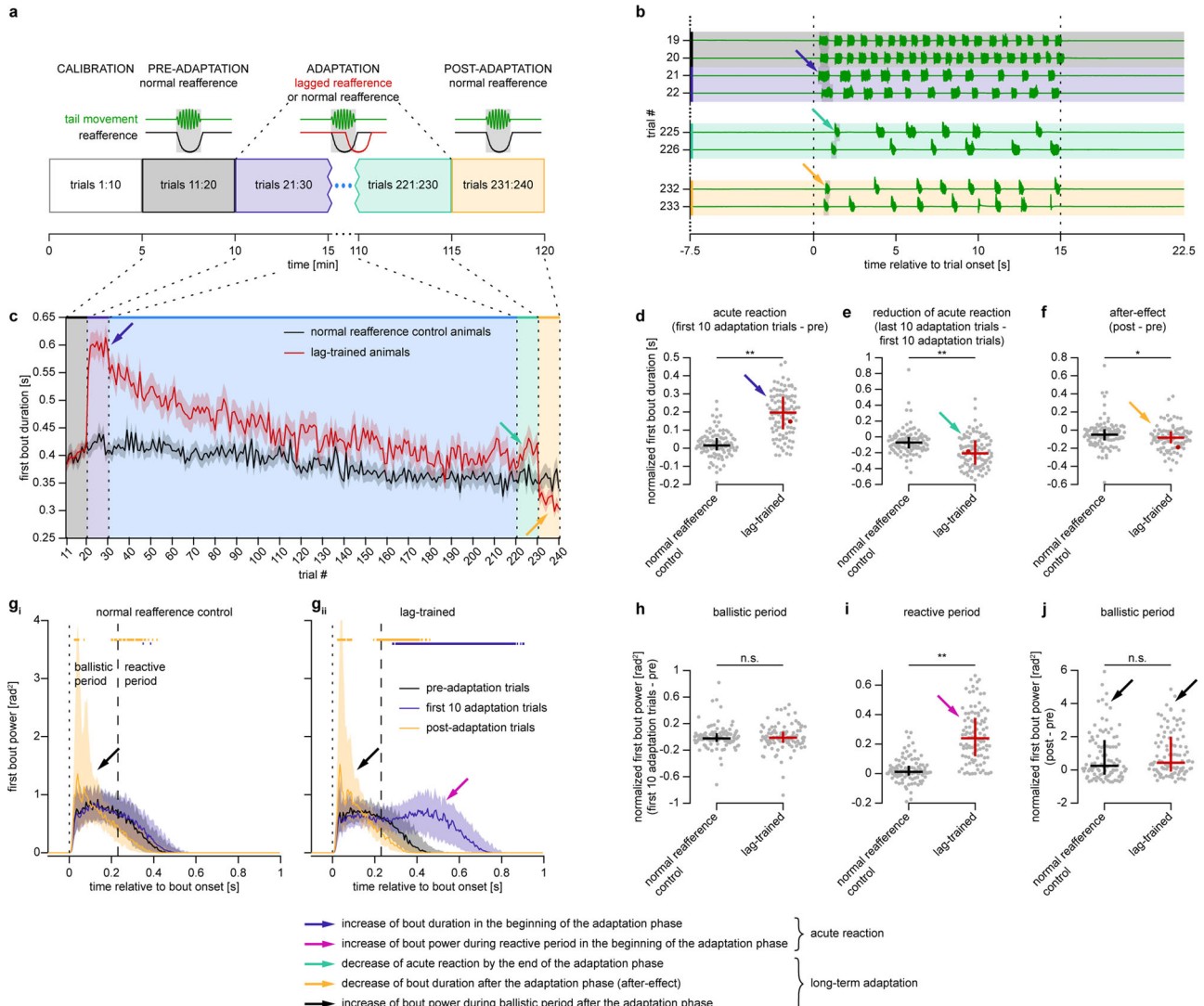

**Fig. 4 Larval zebrafish adapt their behavior in response to a long-lasting perturbation in visual reafference. a** Four phases of the long-term adaptation protocol: calibration, pre-adaptation, adaptation, and post-adaptation. Two groups were tested: normal-reafference control ($N = 103$) and lag-trained larvae ($N = 100$), who received lagged reafference during the adaptation phase (red trace). **b** Tail traces from eight trials in a lag-trained fish. Dotted lines show when the grating was moving forward. Vertical shaded bars indicate first swimming bout in each trial. Blue arrow indicates increase of first bout duration in the beginning of the adaptation phase (acute reaction), cyan arrow indicates decrease of bout duration by the end of the adaptation phase (reduction of acute reaction), and orange arrow indicates decrease of bout duration in the post-adaptation phase (after-effect). **c** First bout duration in each trial (mean ± SEM across larvae). **d–f** Quantification of acute reaction (**d**) and long-term adaptation effects: reduction of acute reaction (**e**) and the after-effect (**f**). Each dot represents first bout duration in one fish, averaged across ten trials and normalized by subtracting the value from the pre-adaptation phase (**d**, **f**) or from the first ten trials of the adaptation phase (**e**). **d–j** Median and interquartile range across larvae are shown, n.s. $p \geq 0.05$, *$p < 0.05$, **$p < 0.01$ (**d** $7.92 \times 10^{-21}$, **e** $1.25 \times 10^{-6}$, **f** 0.02, **h** 0.88, **i** $6.66 \times 10^{-22}$, **j** 0.27; Mann–Whitney $U$-test with two-tailed alternative). Red dots represent example fish shown in **b**. **g** First bout power profile in different experimental phases of normal-reafference control (**i**) and lag-trained larvae (**ii**), averaged within blocks of ten trials in each larva. Dashed lines separate ballistic and reactive periods. Thick horizontal colored lines indicate time points, at which mean bout power is different from the pre-adaptation level (Wilcoxon signed rank test, $p < 0.05/220$, where 220 is the total number of tested time points). Black arrows indicate increase of ballistic power during the experiment. **h–j** Quantification of the change in mean bout power during the experiment: acute reaction of ballistic bout power (**h**), of reactive bout power (**i**), and the after-effect in ballistic bout power (**j**). Each gray dot represents the area below the first bout power curve in one fish, computed over its respective bout period, averaged across ten trials and normalized by subtracting the value from the pre-adaptation phase.

period and a consequent after-effect. In addition, the tail-beat amplitude during the ballistic period is increased during long-term adaptation.

**Long-term adaptation, but not acute reaction, is impaired after PC ablation.** Our hypothesis predicts that long-term adaptation depends on cerebellar output, whilst acute reaction is implemented by a cerebellum-independent mechanism. To test this prediction, we generated a transgenic line expressing

nitroreductase (Ntr) in all cerebellar Purkinje cells (PCs), which allows targeted pharmaco-genetic ablation of PCs by treating the larvae with metronidazole (Fig. 5a; see "Targeted pharmaco-genetic ablation of PCs" section for details). To validate efficiency of the ablation protocol, we acquired confocal stacks of 11 larvae before and after ablation. In all $Ntr^+$ larvae we observed swelling, disappearance and a decrease in the number of PC nuclei, as well as aggregation of the neuropil into puncta with complete loss of the characteristic filiform structure. This demonstrates that even

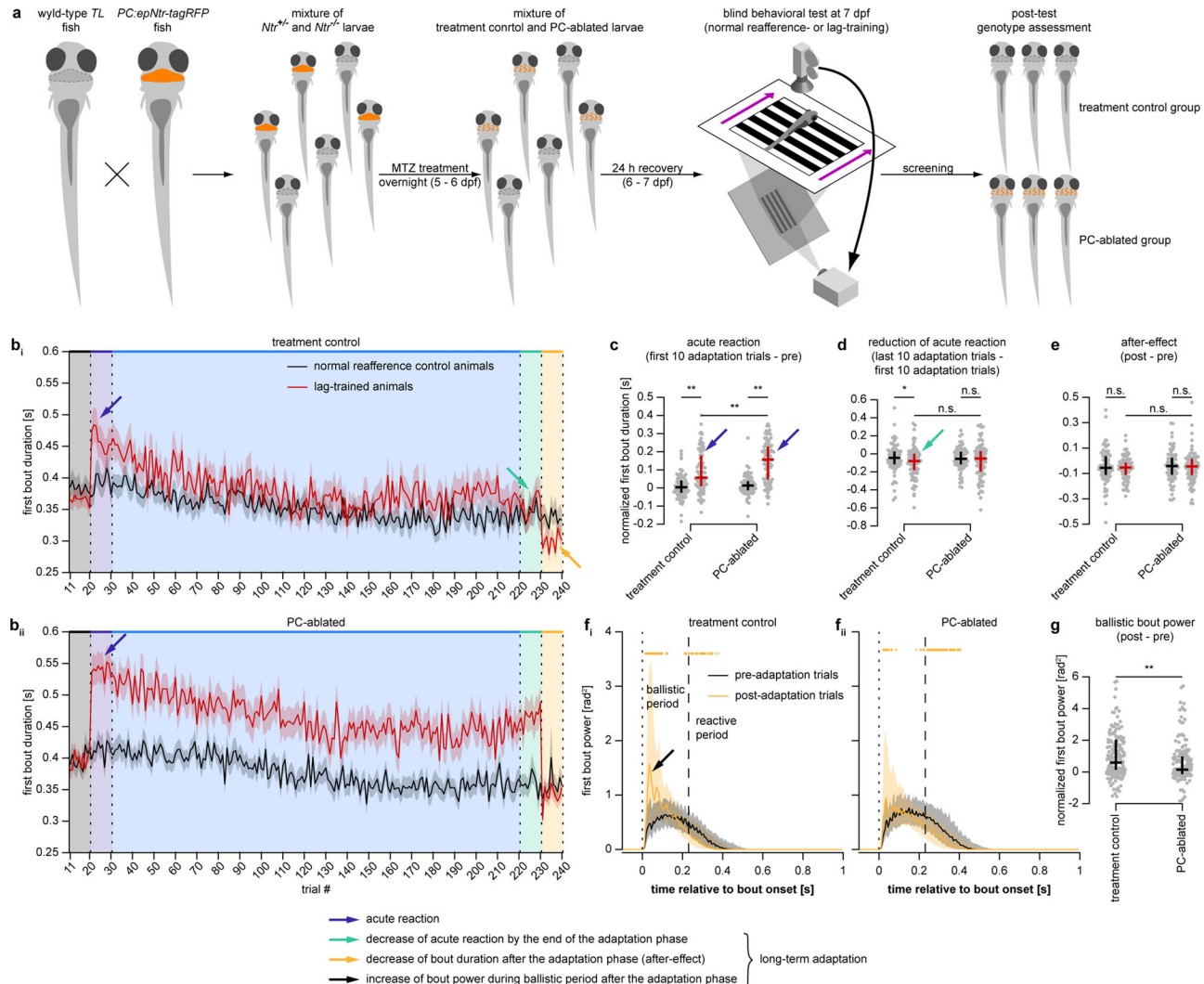

**Fig. 5 Long-term adaptation, but not acute reaction, is impaired after PC ablation. a** Experimental flow of PC ablation experiments. **b** First bout duration in each trial of the long-term adaptation experiment in treatment control larvae (**i** $N = 85$ and 85 for normal-reaference control and lag-trained groups, respectively) and PC-ablated larvae (**ii** $N = 83$ and 90). Solid lines and shaded areas represent mean ± SEM across larvae. **b–e** Blue arrows indicate acute reaction, cyan arrows indicate reduction of acute reaction, and orange arrow indicates the after-effect. **c–e** Quantification of acute reaction (**c**) and long-term adaptation effects: reduction of acute reaction (**d**) and the after-effect (**e**). Each gray dot represents first bout duration in one fish, averaged across ten trials and normalized by subtracting the baseline value obtained during the pre-adaptation phase (**c, e**) or during the first 10 trials of the adaptation phase (**d**). **c–e, g** Black and red lines represent median and interquartile range across larvae. **f**, Mean bout power profiles of the treatment control (**i**) and PC-ablated groups (**ii**). from pre-adaptation and post-adaptation phases of the experiment. First bout power profiles were averaged within respective blocks of ten trials in each larva. Colored curves and shaded areas represent median and interquartile range across larvae. Dotted lines indicate bout onsets, dashed lines separate ballistic and reactive periods. Thick horizontal orange lines indicate time points at which mean bout power during the post-adaptation trials is different from the baseline pre-adaptation bout power (Wilcoxon singed rank test, $p < 0.05/220$, where 220 is the total number of time points, two-tailed alternative). Black arrows indicate increase of ballistic power during the experiment. Data from normal-reaference control and lag-trained animals were pooled together as no effect of reaference condition on increase in ballistic bout power was observed (Fig. 4j). **g** Quantification of the change in mean ballistic bout power during the experiment. Each gray dot represents area below the first bout power curve in one fish, computed over the ballistic period, averaged across ten trials and normalized by subtracting the baseline value obtained during the pre-adaptation phase. **c–e, g** n.s. - $p \geq 0.05$, *$p < 0.05$, **$p < 0.01$ (**c**: treatment control: $8.83 \times 10^{-7}$, PC-ablated: $1.52 \times 10^{-14}$, lag-trained: $0.18 \times 10^{-2}$; **d** treatment control: 0.046, PC-ablated: 0.61, lag-trained: 0.35; **e** treatment control: 0.47, PC-ablated: 0.58, lag-trained: 0.45; **g** $0.68 \times 10^{-2}$; Mann–Whitney U-test with two-tailed alternative).

though some of the PCs remained after ablation, their normal functioning in the circuit was impossible due to destruction of their dendritic trees (Supplementary Fig. 3).

After confirming the efficiency of the ablation protocol, we probed its behavioral consequences. PC-ablated fish were still able to perform the OMR and to acutely react to perturbations in visual reaference. When we probed the responses of PC-ablated zebrafish larvae in the acute reaction paradigm (Fig. 2), we found that responses to all the perturbations (*gain, lag, shunted lag, gain drop*) were still present after

the ablation of PCs (Supplementary Fig. 4). This was also the case when tested in the long-term adaptation paradigm, where naïve PC-ablated larvae were still able to react acutely to unexpected lag in the reaference in the beginning of the adaptation phase (blue arrows in Fig. 5b, c). In fact, the magnitude of the acute reaction to lag was even higher in PC-ablated larvae compared to the treatment control group (Fig. 5c).

On the other hand, long-term adaptation was significantly impaired after PC ablation. The reduction of the acute reaction

(light-green arrows in Fig. 5b, d) was absent in PC-ablated lag-trained animals. Thus, by the end of the adaptation phase, their bouts were still significantly longer than in the PC-ablated normal-reaference control group and longer than their own bouts during the pre-adaptation phase (Fig. 5b). Furthermore, the after-effect was also absent in PC-ablated larvae (Fig. 5b_ii). Although this effect was clearly observed in the treatment control group (orange arrow in Fig. 5b), it was not statistically significant in both groups (Fig. 5e). Finally, the increase in ballistic bout power (black arrow in Fig. 3f) was significantly less prominent in PC-ablated animals compared to the treatment controls (Fig. 5f, g).

Taken together, these results demonstrate that PC ablation does not reduce the acute reaction to unexpectedly perturbed visual reaference but does impair the long-term adaptation to a consistently perturbed reaference condition. This is consistent with the hypothesis that the neuronal controller involved in reactive optomotor swimming does not require an intact cerebellum for its functioning, but that its state can be modulated by a cerebellar internal model.

**Activity of a subpopulation of PCs can represent the output of an internal model.** After observing that the long-term behavioral adaptation to consistently perturbed reaference is a cerebellum-dependent process, we set out to determine whether the output of the cerebellum contained functional features of a recalibrating internal model. To this end, we performed functional imaging of PC activity in zebrafish larvae expressing GCaMP6s in all PCs[27] while they were performing long-term adaptation in a custom-built light-sheet microscope (Fig. 6a). The experimental protocol was modified from the one described in Fig. 4a by shortening the adaptation phase from 210 trials to 50 trials and prolonging the post-adaptation phase from 10 trials to 50 trials (Fig. 6b). This was done to ensure stable recordings of the PC activity during the whole experiment, and to follow the cell activity for a longer time after the adaptation.

We first divided lag-trained larvae into two groups referred to as adapting and non-adapting (Fig. 6c) based on the magnitude of the reduction of acute reaction during the adaptation phase. Larvae that decreased their bout duration during the adaptation phase by at least 40 ms were considered adapting. We verified that despite the small sample size (8, 8, and 9 larvae in the normal-reaference control, lag-trained non-adapting and lag-trained adapting groups, respectively), the shorter adaptation phase and the microscope excitation light, the long-term adaptation effects were still detectable (Supplementary Fig. 5). Lag-trained adapting larvae acutely reacted to presentation of lagged reaference in the beginning of the adaptation phase (blue arrows in Supplementary Fig. 5b_iii, c_i), reduced the magnitude of the acute reaction by the end of the adaptation phase (light-green arrows in Supplementary Fig. 5b_iii, c_ii), and demonstrated a clear after-effect in the beginning of the post adaptation phase (orange arrow in Supplementary Fig. 5b_iii, c_iii). It is important to note that non-adapting lag-trained animals not only failed to exhibit long-term adaptation (Supplementary Fig. 5b_ii, c_ii-iii), but also showed a barely detectable acute reaction (Supplementary Fig. 5b_ii, c_i). Therefore, the more correct label for this group would be "not reacting to changes in visual feedback". We use the term "non-adapting" for convenience.

After confirming that the long-term adaptation effects were detectable in lag-trained adapting larvae and not in other experimental groups, we turned to analyzing the activity of ROIs in the cerebellum ($N = 366 ± 19$ ROIs, mean ± SEM across all 25 imaged larvae; see "PC functional imaging data analysis" section for details). The activity of two example ROIs in several trials sampled from different phases of the experiment is presented in

Fig. 6d_i, and their location within the cerebellum is presented in Fig. 6a. We first computed the first-bout-triggered response in each trial by averaging the bout-triggered fluorescence of each ROI (Fig. 6d_ii) in a 1.2 s window after the first bout onset (Fig. 6d_iii). Then, we measured how much the bout-triggered response changed during four crucial transitions in the experimental protocol by computing the following four criteria for each ROI (Fig. 6d_iv):

1. Criterion 1: how much the bout-triggered response increased in response to unexpected presentation of lagged reaference to a naïve larva.
2. Criterion 2: how much the response increased during the adaptation phase, while the lag-trained animals were adapting to a novel reaference condition.
3. Criterion 3: how much the response increased when the reaference condition was switched back to normal.
4. Criterion 4: how much the response increased during the post-adaptation phase, while the animals were adapting back to the original reaference condition.

We used these criteria to assign a 4-digit ternary barcode to each ROI (Fig. 6d_v; see "PC functional imaging data analysis" section). Each digit corresponds to one of the criteria and can take one of three values: "+" (increase of bout-triggered response), "−" (decrease), or "0" (no change). Thus, for example, barcode 0-0+ assigned to ROI 1 in Fig. 6d means that this ROI did not change its bout-triggered response in the beginning of the adaptation phase, decreased the response during the adaptation phase, did not change the response in the beginning of the post-adaptation phase, and increased the response back during the post-adaptation phase. In contrast, ROI 2 was assigned with a barcode +0−, as it increased its response when lagged reaference was first introduced, did not significantly change during the adaptation phase, and decreased the response when the reaference condition was set back to normal and during the rest of the post-adaptation phase.

We aimed to find activity profiles that were significantly enriched in lag-trained adapting larvae compared to the other two groups, because these activity profiles might reflect the output of a recalibrating internal model. To this end, we divided all ROIs into clusters based on their barcodes. We found that the only cluster that contained significantly higher fractions of ROIs in lag-trained adapting fish was the 0-0+ cluster ($9.3 ± 5.4\%$ ($33.8 ± 19.2$ out of $334.2 ± 28.2$ ROIs) in lag-trained adapting fish, $0.2 ± 0.2\%$ ($1.0 ± 0.9$ out of $386.3 ± 40.1$ ROIs) in lag-trained non-adapting fish and $0.9 ± 0.3\%$ ($3.6 ± 1.5$ out of $381.8 ± 33.3$ ROIs) in normal-reaference control fish, mean ± SEM across larvae; Fig. 6e). ROI 1 from Fig. 6d belongs to this cluster.

The bout-triggered responses of these ROIs gradually decreased during the adaptation phase (negative criterion 2) and increased back to the original level during the post-adaptation phase (positive criterion 4) (Fig. 6f). Such an activity profile is similar to the expected output of a putative internal model, which monitors the motor-to-sensory transformation rule and gradually recalibrates if a long-lasting and therefore learnable change in this rule occurs.

To test whether activity of the 0-0+ ROIs could be explained simply by motor activity of the larvae, which was different across groups by design, we modeled an artificial ROI, which linearly encodes motor activity (motor regressor) for each fish and processed it in exactly the same way as we processed the real ROIs (Supplementary Fig. 6a). As expected, bout-triggered responses of the motor regressors were markedly similar to the behavior of the larvae (Supplementary Fig. 6b, c), showing an acute increase in response to unexpected presentation of lagged reaference (dark-blue arrows in Supplementary Fig. 6a–c). However, this acute

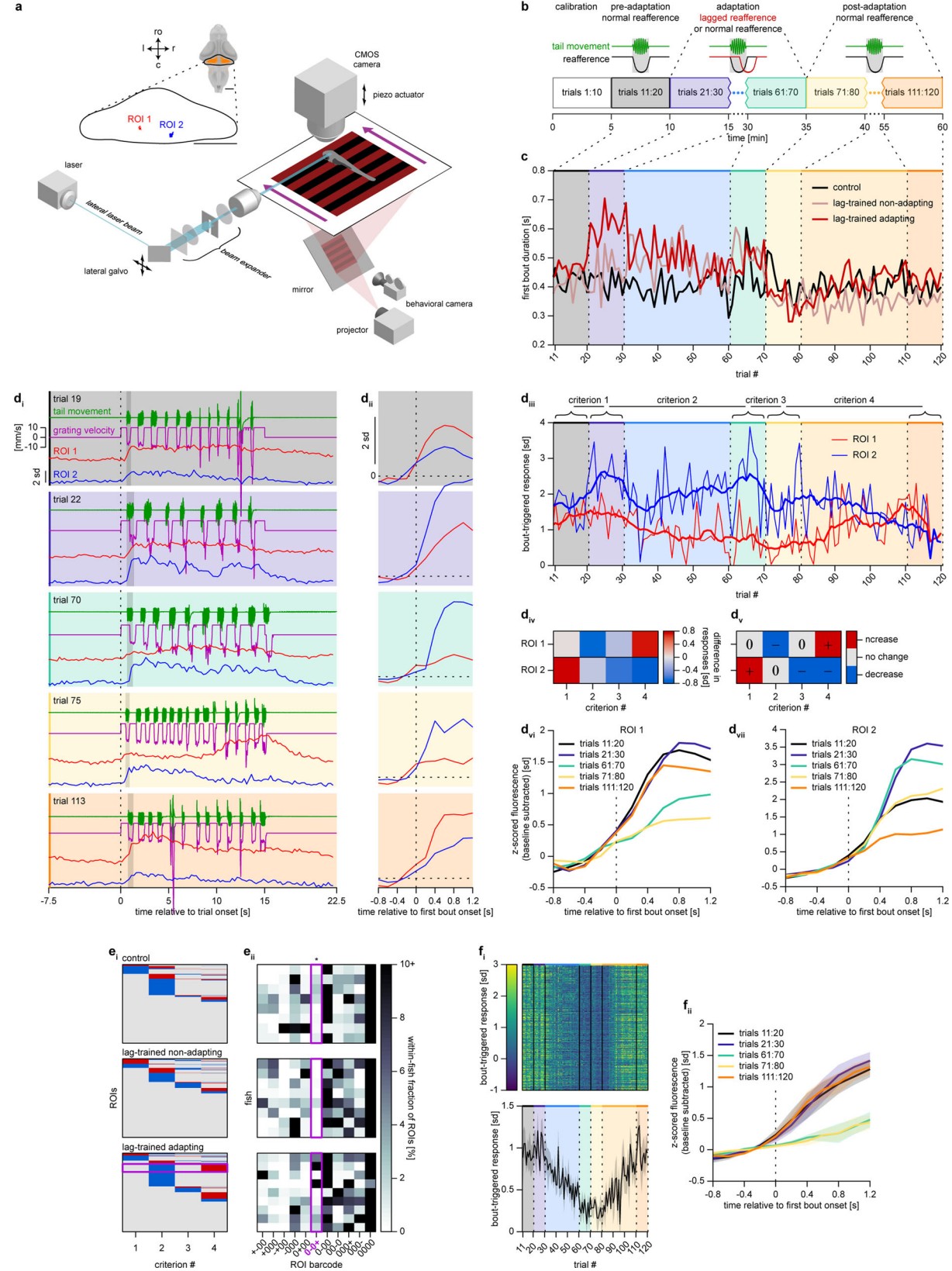

reaction was not observed in response profiles of $0\text{-}0+$ ROIs (Supplementary Fig. 6d). Moreover, we were able to find ROIs whose activity actually showed an acute reaction and was in general similar to the motor regressor (Supplementary Fig. 6e, see also activity profile of the ROI 2 shown in Fig. 6d in blue. This ROI is an example of such motor-related ROIs). Therefore, the

observed dynamics of bout-triggered responses of $0\text{-}0+$ ROIs cannot be explained by a motor component.

ROIs belonging to this functional cluster were located predominantly in the medial cerebellum of lag-trained adapting larvae (Supplementary Fig. 7—top row). However, when we randomly sampled the same number of ROIs, we observed a

**Fig. 6 Activity of a subpopulation of PCs can represent the output of an internal model. a** Light-sheet microscope combined with a behavioral rig used for PC imaging. Inset shows location of two example ROIs shown in **d**. Scale bar: 100 µm; ro rostral direction, l left, r right, c caudal. **b** Modified long-term adaptation protocol. **c** First bout duration in each trial in normal-reaference control larvae ($N = 8$), lag-trained non-adapting larvae ($N = 8$), and lag-trained adapting larvae ($N = 9$), mean across larvae is shown. **d** Imaging data processing flow shown for two example ROIs (see details in text). **d_i** Z-scored fluorescence in five trials. Vertical shaded bars indicate first swimming bout in each trial. **d_ii** First-bout-triggered fluorescence. **d_iii** First-bout-triggered responses in each trial. Thick lines represent box-filtered responses, with a filter length of nine trials. **d_iv** Four criteria that represent change in bout-triggered responses during important transitions of the experimental protocol. **d_v** Criteria converted into 4-digit ternary barcodes. **d_vi** First-bout-triggered fluorescence averaged across respective blocks of ten trials. **e** Clustering of ROIs using barcodes. **e_i** Barcodes of all ROIs pooled from imaged larvae. **e_ii** Within-fish fractions of ROIs assigned to different clusters. Only clusters containing, on average, at least 2% of ROIs in at least one experimental group are shown. Magenta rectangles indicate 0-0 + cluster, which was the only cluster that was significantly enriched in lag-trained adapting larvae (*$p = 0.011$; Kruskal–Wallis test). **f_i** (top), First-bout-triggered responses of all 0-0 + ROIs pooled from lag-trained adapting larvae. **f_i** (bottom), First-bout-triggered responses of 0-0 + ROIs, averaged within each lag-trained adapting larva. Solid line and shaded area represent mean ± SEM across larvae. **f_ii** First-bout-triggered fluorescence of 0-0 + ROIs, averaged across respective blocks of ten trials and across ROIs within each lag-trained adapting larva. Colored lines and shaded areas represent mean ± SEM across larvae.

qualitatively similar spatial distribution (Supplementary Fig. 7—bottom row) suggesting that 0-0+ ROIs represent a distributed population of PCs without apparent spatial organization.

In summary, we have found that the only type of PCs that was significantly enriched in lag-trained adapting larvae was 0-0+. These ROIs gradually decreased their bout-triggered activity while the animals were adapting to a novel reaference condition and gradually returned back to baseline when the condition was switched back to normal. This activity profile cannot be explained by motor activity of behaving larvae and rather represents the output of a recalibrating internal model of the motor-to-sensory transformation rule.

**Long-term adaptation is accompanied by a decrease in time constants of sensory integration.** In our last experiment, we aimed to test if ROIs with functional properties of internal models also exist outside the cerebellum. To achieve this goal, we imaged whole-brain activity of animals performing long-term adaptation and performed the same analysis as in Fig. 6 (computing 4-digit barcodes). We found only insignificant fraction of 0-0+ ROIs without any apparent spatial organization in all the experimental groups (normal-reaference control, $N = 12$, lag-trained non-adapting, $N = 13$, and lag-trained adapting, $N = 6$). It is important to note that the transgenic line that we use for whole-brain imaging has very poor expression in PC layer, which explains lack of 0-0+ ROIs identified in this experiment. In line with results of the ablation experiment (Fig. 5), this suggests that the major role in long-term adaptation belongs to the cerebellum.

We then aimed to understand the effects that a recalibrating internal model may have on the feedback controller involved in OMR. It was proposed that one action of the cerebellum is to modify the time constant of the oculomotor neuronal integrator in the brainstem during eye movements[28,29]. We therefore tested if, similarly, the time constants of the optomotor integrator were modified during long-term adaptation. Just as we did in Fig. 3, we identified sensory and motor ROIs, and estimated time constants of all sensory ROIs in blocks of ten trials. We then computed the difference between time constants in the end and the beginning of the adaptation phase (Fig. 7a). We observed that lag-trained larvae had fewer sensory ROIs that increased their time constant compared to the normal-reaference control group, and more ROIs that decreased their time constant compared to the other two groups. Therefore, successful long-term adaptation appears to be associated with a decrease in time constants of a subpopulation of sensory ROIs. These ROIs with decreasing time constants were evenly distributed across brain regions involved in sensory integration identified in Fig. 3 (the dorsal raphe with its surrounding reticular formation and diencephalic regions such as the pretectum) (Fig. 7b).

## Discussion

In this study, we propose that the OMR and its acute reactions to unexpected perturbations in reaference are implemented by a feedback control mechanism. This contradicts the previously proposed internal-model-based mechanism suggesting that fish use an internal representation of expected reaference and adapt their behavior if the actual reaference does not meet this expectation[21–23]. In those studies, the behavioral changes in response to perturbed reaference were termed "adaptive loco-motion", highlighting that larvae actually adapt their behavior when they experience a sudden unexpected perturbation. The term "adaptation" implies that something in the brain has changed during this process. Here, we use the term "reactive locomotion" to emphasize that we believe these behavioral changes represent reactions of a simple feedback controller without any changes in the underlying circuitry.

The proposed mechanism represents a straightforward implementation of the OMR definition in a control loop. The OMR therefore stabilizes the animal's position with respect to its visual environment using a feedback controller that tries to keep the integrated optic flow at zero. We were surprised to find that implementing this simple definition in a circuit not only results in swimming behavior similar to that of real larvae (compare Supplementary Fig. 1a with Fig. 1d) but also closely reproduces the acute reactions to all tested perturbations in reaference (Fig. 2c, d). The model suggests that bout and interbout duration is determined by the interplay between the two variables computed by the controller: sensory drive and "tiredness". Sensory drive accumulates while the grating moves forward and can be interpreted as motivation to swim. In the model, an increased sensory drive leads to longer bouts and shorter interbouts. On the other hand, larval zebrafish rarely perform bouts that are longer than 500 ms. This is implemented in the model by a motor integrator, whose output can terminate the bouts and prolong subsequent interbouts. For convenience, we refer to the output of the motor integrator as a metric of "tiredness" although it encapsulates multiple reasons for bout termination.

Although the model is clearly a simplification, we believe that its core principles represent the sensory-motor transformation underlying control of bout and interbout duration because this interplay between sensory drive and "tiredness" can explain all the subtleties in observed acute reactions. Thus, high gains decrease sensory drive during bouts which results in shorter bouts and longer interbouts (Fig. 2c_i–d_i). On the other hand, very low gains result in increased sensory drive, which significantly pro-longs the bouts (Fig. 2c_i). If the bouts are very long, the effect of high "tiredness" starts to dominate over high sensory drive, and as a result, the interbouts following these long bouts would be longer despite high sensory drive. This explains the peculiar

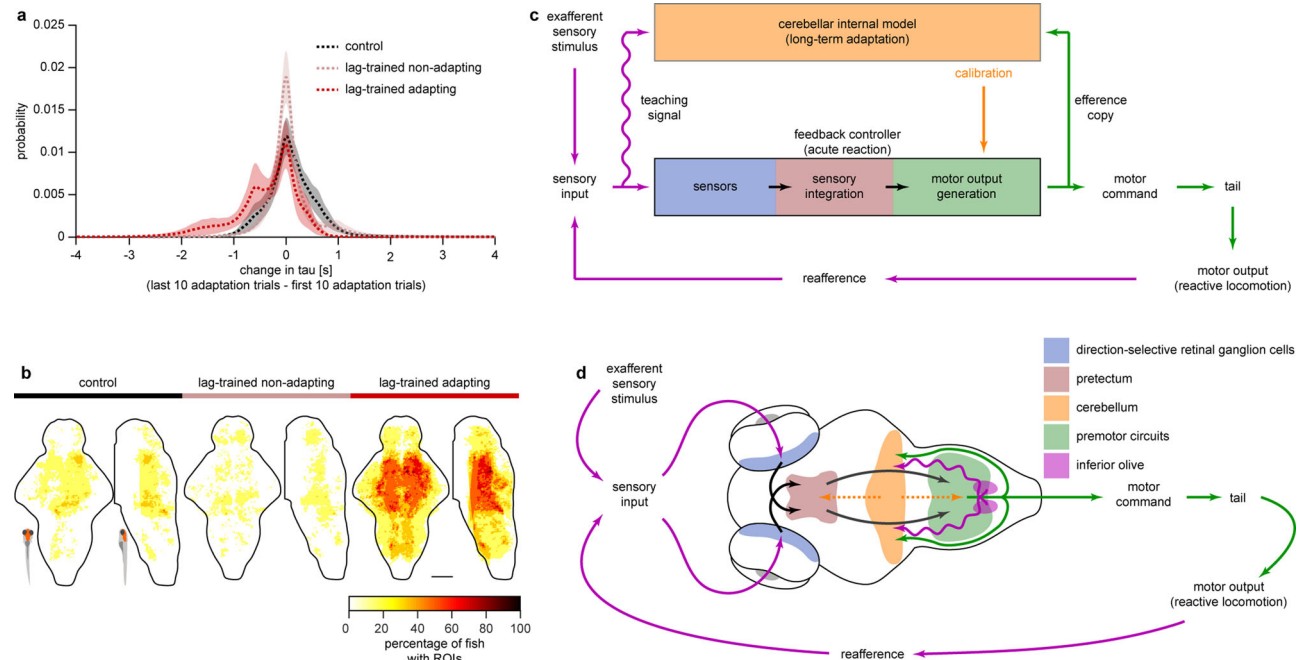

**Fig. 7 A cerebellar internal model calibrates a feedback controller involved in sensorimotor control. a** Probability distribution of differences in time constants of sensory ROIs (end of adaptation—beginning of the adaptation). Histograms were smoothed using a kernel function with width of 0.1 s. Dotted lines denote mean probability values, shaded areas denote SEM across larvae (normal-reafference control, $N = 12$, lag-trained non-adapting, $N = 13$, and lag-trained adapting, $N = 6$). **b** Anatomical location of sensory ROIs that decreased their time constants by more than 0.4 s (see Supplementary Fig. 2 for anatomical reference). **c** Schematic diagram of the feedback controller that can implement acute reaction to unexpected perturbations in reafference. Cerebellar internal model monitors the efference copies of motor commands and resulting sensory consequences and learns their transfer function. It calibrates some intrinsic parameters of the controller according to consistent environmental features that can be learned by an internal model. The wavy line denotes the teaching signal used by the internal model to learn the transfer function. The dotted orange arrows denote potential influence of the cerebellum over the feedback controller (modification of time constants of sensory integration, as suggested by **a**, and/or parameters of the pre-motor circuits). **d** Mapping of the crucial functional nodes involved in acute reaction and long-term adaptation onto the larval zebrafish brain. See details in the text.

V-shaped trend in interbout duration as a function of gain (Fig. 2d_i).

In the case of lagged reafference, bouts are prolonged due to increased sensory drive in the beginning of the bout when there is no reafference (i.e., no slowing down of the grating) (Fig. 2c_ii–c_iii). In addition, the model explains why the interbouts are prolonged only in the non-shunted lag setting but not in the shunted one (Fig. 2d_ii–d_iii). In the non-shunted setting, the grating continues to move backwards after the bout offset, so the sensory drives continues to decrease, which leads to prolonged interbouts (Fig. 2d_ii). In the shunted setting, however, the grating returns to forward motion immediately after the bout offset, so the interbouts are not prolonged (Fig. 2d_iii).

Finally, the model explains why perturbations in reafference during early bout segments have more influence over the bout duration, whereas perturbations late in the bout affect the subsequent interbout more (Fig. 2c_iv–d_iv). The reason is that the influence of "tiredness" and sensory drive over the final motor output changes throughout the course of a bout. Since the "tiredness" accumulates during the bout, it starts to affect the output more strongly and to dominate over the sensory drive by the end of the bout. Simply, if the fish is tired, it does not matter how much swimming drive it may have. As a result, in the beginning of the bout, while the fish is not yet tired, an increase in sensory drive effectively prolongs the bout. If, however, sensory drive is increased late in the bout, it does not prolong the bout as much because by then, the bout duration is almost completely determined by the increasing "tiredness". In this case, the "tiredness" by the end of the bout will be less than if the

reafference was perturbed early in the bout, so the interbout duration will decrease more.

In order for this theoretical mechanism to work in real larvae, they must be able to integrate the optic flow to compute the sensory drive. Our whole-brain functional imaging experiments revealed that the process of sensory integration of the forward visual motion indeed takes place in several brain regions including the pretectum (Fig. 3j and Supplementary Fig. 2). An increasing body of work suggests that the pretectum plays a crucial role in whole-field visual processing and visuomotor behaviors in larval zebrafish[30–34]. It has been shown that pretectal neurons integrate monocular direction-selective inputs from the two eyes and drive activity in the premotor hindbrain and midbrain areas during optomotor behavior[31]. Together with recent evidence from different experimental paradigms[25,26], the present study demonstrates that the pretectum is involved not only in the binocular integration of sensory inputs, but also in temporal integration that can underlie accumulation of the sensory drive.

The proposed feedback control mechanism of reactive optomotor locomotion can therefore be mapped onto the larval zebrafish brain (Fig. 7c, d). The sensory part of the feedback controller starts with direction-selective velocity sensors. Direction-selectivity can be observed in the brain already at the level of retinal ganglion cells in vertebrates[35,36], including zebrafish larvae[37,38]. The velocity integrator can correspond to the pretectum, which receives projections from the contralateral direction-selective retinal ganglion cells[31,39,40]. Finally, the pretectal neurons (velocity integrators) send anatomical projections to premotor areas in the hindbrain and to the nucleus of the

medial longitudinal fascicle[33,41,42]. As these regions both displayed motor-related activity in our experiments (Fig. 3f and Supplementary Fig. 2), we hypothesize that they correspond to the premotor parts of the controller. In essence, this study suggests that the major role of the pretectum in visuomotor behaviors is computing the sensory drive, or motivation, to perform a motor action[34]. This predicts that motor output of the fish should increase in response to increased activity of pretectal neurons, and vice versa, although more studies would be required to address this prediction.

Finally, our loss-of-function experiments have demonstrated that, in contrast with previous predictions[21–23], acute reaction was not reduced after ablation of the PCs (Fig. 5b, c and Supplementary Fig. 4). Consequently, the neuronal controller involved in reactive optomotor behavior does not require intact cerebellum for its functioning.

In the present study, we demonstrate that larval zebrafish are able to gradually adapt their behavior in response to a long-lasting perturbation in reafference (Fig. 4), and that this process is cerebellum-dependent (Fig. 5). This is consistent with the view of the cerebellum as a neuronal substrate of internal models[8,12–16,43].

The notion that the cerebellum is not involved in online corrections of the movements in response to wrong sensory feedback, but is involved in learning relations between movements and their feedback in order to adapt the behavior in a predictive manner, has been already proposed in the cerebellar literature[44]. In two studies[15,16], humans with impaired cerebellar function were able to update the motor program during a reaching task after the feedback of their movements was perturbed by the experimenters. However, they were not able to update their feedback estimation after the adaptation session, indicating that the cerebellum is involved in acquiring and updating a forward internal model. In our long-term adaptation experiments, a similar process has taken place. During long-lasting exposure to consistently perturbed reafference, animals with an intact cerebellum may have recalibrated their forward models to reduce their expectations, making them consistent with the new environmental condition.

In this study, we took advantage of the zebrafish larvae's accessibility for functional imaging to monitor this process of recalibration in the cerebellum during motor learning. We have identified a subpopulation of PCs that gradually decreased their responses while the animals were adapting to insufficient reafference, and increased their responses back to the original level while the animals were getting used to the original condition after the adaptation. This activity profile is similar to the expected output of a recalibrating internal model, with the bout-triggered activity in this population corresponding to a putative "expected visual feedback" signal.

Although this study provides direct evidence of internal models in the cerebellum, the exact processes that take place in the cerebellum of adapting animals remain unclear. Since functional imaging provides only a coarse view about how motor, sensory, and expectation-related signals are represented in PCs, future electrophysiological investigations are required to address these questions. Nevertheless, the fact that we observed sensory-related activity in the inferior olive (Fig. 3f, j) indirectly suggests that the cerebellum in larval zebrafish acts as a forward model during the OMR, and the highly sensory nature of PCs' complex spikes that directly result from action potentials in the inferior olive[45] was also reported recently[27]. Inferior olive activity is believed to convey a teaching signal to the PCs[46,47] that updates the internal models in the cerebellum[9,48] by modifying synaptic weights in the cerebellar circuitry[49,50]. Since the teaching signal must be expressed in the same coordinates as the output of the internal

model, the sensory nature of the teaching signal suggests that the cerebellar internal model involved in our experiments is forward in nature, although this distinction is likely to be more subtle.

However, in a recent study[43], also conducted on larval zebrafish performing OMR in response to a moving grating, calcium activity of PCs was related to predicting the next trial. This suggests that in a different experimental setting, cerebellum may be involved in encoding an internal model of the environment rather than of interaction between oneself and the environment, consistently with older work in cats[51]. This seeming contradiction reinforces the view of the cerebellum as a neuronal learning machine[52] whose beautiful architecture enables it to model an inconceivable number of patterns in an inconceivable variety of contexts[53].

One interesting result of this study is that bout power during the initial ballistic period of the bouts did not depend on the current reafference condition: the magnitude of the first several tail oscillations is fixed regardless of what sensory input the fish receives (Fig. 2e). This means that the behavior during this ballistic period is pre-determined by the state of the neuronal controller that converts information about moving grating into optomotor behavior. However, if larval zebrafish are exposed to a long-lasting perturbation in reafference, the magnitude of the initial tail oscillations increases (Fig. 4g), and we show that this process is cerebellum-dependent (Fig. 5f, g). This means that the cerebellum can influence the state of the neuronal controller of the OMR, and if the internal model in the cerebellum updates, the state of the controller updates as well (Fig. 7c, d).

Unexpectedly, the increase in ballistic bout power was observed not only in the lag-trained group but also in the normal-reafference control group, which never experienced perturbed reafference (Fig. 4j). This can be explained by the fact that what we call the normal reafference condition in the closed-loop experiments may in fact differ from the real sensory feedback that larval zebrafish perceive upon free unrestrained swimming bouts. For example, the reafference presented by the projector always lags slightly behind the tail movements due to the software processing delay. In addition, the way by which we computed the swimming velocity from the tail movements may only approximate the real velocity that the fish would have reached if it were freely swimming. Finally, in our closed-loop experiments, head-restrained larvae did not receive any vestibular and lateral line sensory feedback. Therefore, even though the normal reafference condition was calibrated to closely match the visual real-life conditions, it was inevitably different from the total sensory feedback received by freely swimming larvae. Future experiments involving translation of the long-term adaptation experiment to the freely-swimming environment would be required to identify whether this disparity underlies observed recalibration of the neuronal controller of the OMR.

The exact pathway by which the cerebellum affects the circuitry involved in OMR remains unclear. One possibility is that it acts upon the premotor parts of the circuit, as cerebellar projection neurons send their output to the nucleus of the medial longitudinal fascicle and to the hindbrain[42]. In this scenario, the cerebellum would recalibrate the motor parts of the controller during the long-term motor adaptation, so that fish would behave differently in response to the same sensory drive. Another possibility is that the cerebellum recalibrates the sensory parts of the OMR circuitry during the adaptation. It was proposed for the case of the eye movements that the action of the cerebellum is to modify the time constant of the oculomotor neuronal integrator in the brainstem[28,29]. Similarly, in the case of the OMR adaptation, the role of the cerebellar internal model might be to fine-tune the time constant of the optic flow integrator in the pretectum. In this study, long-term adaptation to perturbed visual

feedback was associated with a change in integration time constants of sensory ROIs (Fig. 7a), demonstrating plausibility of this hypothesis. This influence can be mediated by anatomical projections from the cerebellum to the pretectum, which have been described in zebrafish[54]. More generally, it may be that the recalibration does not in fact happen at a unique point in the circuitry, but that the cerebellum is able to undergo synaptic changes that support the homeostatic recalibration of function within a circuit, after this has been driven away from its stable point by perturbations in reafference[55].

As mentioned in the introduction, stimulus-driven behaviors can be naturally understood in the framework of feedback controllers, where the behavior results in cancellation of the stimulus that evokes it. It is natural to believe that though our study pertains to the OMR in larval zebrafish, the principle may very well extend to other stimulus-driven behaviors and even to more abstract yet predictable relationships between behavior and the environment[56], such as situations involving reward[57] or social interactions[58].

In conclusion, our results demonstrate the role of cerebellar internal models in calibrating the neuronal circuits involved in reactive motor control. This process ensures that the circuit is well-tuned in the sense that the animal's behavior has the required effect on its sensory environment.

## Methods

### Experimental model

*Zebrafish husbandry.* All experiments were conducted on larval zebrafish (*Danio rerio*) at 6–8 days of post-fertilization (dpf) of yet undetermined sex. All animal procedures were performed in accordance with approved protocols set by the Max Planck Society and the Regierung von Oberbayern (Protocol number 55-2-1-54-2532-82-2016).

Both adult fish and larvae were maintained at 28 °C on a 14/10 h light/dark cycle, unless otherwise specified. Adult zebrafish were housed in a zebrafish facility system with constantly recirculating water with a daily 10% exchange. The system fish water was deionized and adjusted with synthetic salt mixture (Instant Ocean) to 600 μS conductivity, with the pH value adjusted to 7.2 using NaHCO₃ buffer solution. The water was filtered over bio-filters, fine-filters, and carbon filters and UV-treated during recirculation. Adult zebrafish were fed twice a day with a mixture of Artemia and flake feed.

To obtain larvae for experiments, one male and one female (in some cases, three male and three female) adult zebrafish were placed in a mating box in the afternoon and kept there overnight. The embryos were collected in the following morning and placed in an incubator that was set to maintain the above light and temperature conditions (Binder, Germany). Embryos and larvae were kept in 94 mm Petri dishes at a density of 20 animals per dish in Danieau's buffer solution (58 mM NaCl, 0.7 mM KCl, 0.4 mM MgSO₄, 0.6 mM Ca(NO3)₂, and 5 mM HEPES buffer) until 1 dpf and in fish water from 1 dpf onwards. The water in the dish was changed daily.

*Zebrafish strains.* Purely behavioral experiments were conducted using wild-type Tupfel long-fin (*TL*) zebrafish strain or transgenic *Tg(PC:epNtr-tagRFP)* line that was used for PC ablation (see below).

Efficiency of PC ablation was evaluated using the progeny of *Tg(PC:epNtr-tagRFP)* zebrafish outcrossed to fish expressing GCaMP6s in PC nuclei and RFP in PC somata (*Tg(Fyn-tagRFP:PC:NLS-GCaMP6s)*)[27]. This allowed evaluating effects of the ablation protocol on the morphology of both cell nuclei and somata. These larvae were homozygous for *nacre* mutation, which introduces a deficiency in *mitfa* gene that is involved in development of melanophores[59]. As a result, homozygous *nacre* mutants lack optically impermeable pigmented spots on the skin, which enables brain imaging without invasive preparations.

Whole-brain functional imaging experiments were conducted using transgenic zebrafish strain with pan-neuronal expression of GCaMP6s (*Tg(elavl3:GCaMP6s)*)[24]. PC functional imaging experiments were conducted using zebrafish that expressed GCaMP6s specifically in PCs (*Tg(PC:-GCaMP6s)*)[27]. In both cases, the animals were also homozygous for *nacre* mutation.

A Z-stack of larval zebrafish reference brain used for anatomical registration of the whole-brain functional imaging data was previously acquired in our laboratory by co-registration of 23 confocal z-stacks of zebrafish brains with pan-neuronal expression of GCaMP6f (*Tg(elavl3:GCaMP6f)*)[60], homozygous for *nacre* mutation. For anatomical registration of PC functional imaging data, the red channel of one confocal stack of *Tg(Fyn-tagRFP:PC:NLS-GCaMP6s)* was used as a reference.

*Targeted pharmaco-genetic ablation of PCs.* To perform targeted ablation of PCs, we employed Ntr/MTZ pharmaco-genetic approach that has been successfully used

in zebrafish[61–63]. This method is based on treating the animals expressing nitroreductase (Ntr) in a cell population of interest with prodrug metronidazole (MTZ). Ntr converts MTZ into a cytotoxic DNA cross-linking agent leading to death of cells of interest. To this end, we generated a transgenic line that expressed enhanced Ntr (epNtr)[63] under the PC-specific *carbonic anhydrase 8* (*ca8*) enhancer element[64]. epNtr fused to tagRFP was cloned downstream to the aforementioned enhancer and a basal promoter. This construct (abbreviated as *PC:epNtr-tagRFP*) was injected into nuclei of single cell stage *TL* embryos heterozygous for *nacre* mutation, at a final concentration of 20 ng/μl together with 25 ng/μl *tol2* mRNA. Larvae showing strong RFP expression in PCs were raised to adulthood as founders and outcrossed to *TL* fish to gain a stable line.

Ablation-induced changes in behavior were tested using the progeny of a single founder. The embryos obtained from a *PC:epNtr-tagRFP*⁺/⁻ fish outcrossed to a *TL* fish were screened for red fluorescence in the cerebellum at 5 dpf, and 10 RFP-positive (*PC:epNtr-tagRFP*⁺/⁻) and 10 RFP-negative (*PC:epNtr-tagRFP*⁻/⁻) larvae were kept in the same Petri dish to ensure subsequent independent sampling. At 18:00, most of the water in the dish was replaced with 10 mM MTZ solution in fish water, and larvae were incubated in this solution overnight in darkness for 15 h. The next morning at 9:00, animals were allowed to recover in fresh fish water. The next day, behavior of 7 dpf MTZ-treated larvae was tested in a respective behavioral protocol. After the experiment, the animals were screened for red fluorescence once again to reassess their genotype after mixing positive and negative larvae in one Petri dish. *PC:epNtr-tagRFP*⁻/⁻ and *PC:epNtr-tagRFP*⁺/⁻ siblings constituted treatment control and PC ablation experimental groups, respectively.

Efficiency of the PC ablation protocol was evaluated using progeny of *Tg(PC:epNtr-tagRFP)* fish outcrossed to *Tg(Fyn-tagRFP:PC:NLS-GCaMP6s)* fish, homozygous for *nacre* mutation. These larvae underwent the same ablation protocol, and z-stacks of their cerebella in RFP and GFP channels were acquired under the confocal microscope (LSM 700, Carl Zeiss, Germany) before and after the ablation (at 5 and 7 dpf, respectively). To quantify structural segregation of the PCs induced by ablation, we first masked the cerebellum in each confocal stack using the pipra software[65]. Next, we computed the local entropy as a non-reference metric of tissue inhomogeneity[66]. It was computed across the whole stack in radii of 7 μm using the entropy implementation in scikit-image[67]. We then averaged all entropy values inside of the cerebellar masks to gain an entropy estimate in bits for each stack. We hypothesized that maintaining the anatomical structure suggests a high entropy, whereas a structural collapse caused by ablation decreases the entropy. As expected, we observed that *Ntr*⁺ fish displayed lower entropy values within the cerebellum compared to *Ntr*⁻ control fish after metronidazole treatment (Supplementary Fig. 3d).

### Closed-loop experimental assay in head-restrained zebrafish larvae.

All experiments were conducted using head-restrained preparations of 6–8 dpf zebrafish larvae, similar to ref. [21]. For behavioral experiments, each larva was embedded in 1.5% low melting point agarose (Invitrogen, Thermo Fisher Scientific, USA) in a 35 mm Petri dish. For functional imaging experiments, larvae were embedded in 2.5% agarose in custom 3D-printed plastic chambers (Form 2, Standard Clear Resin V4, Formlabs, USA), with glass coverslips sealed on the front and left sides of the chamber using grease, at the entry points of the frontal and lateral laser excitation beams, and the agarose around the head was removed with a scalpel to reduce scattering of the beams (see "Light-sheet microscopy" section). After allowing the agarose to set, the dish/chamber was filled with fish water and the agarose around the tail was removed to enable unrestrained tail movements that were subsequently used as behavioral readout.

A dish/chamber with an embedded larva was then placed onto the screen of the custom-built behavioral or functional imaging rig (Figs. 1c, 3a, and 6a). In the behavioral rig, the screen with the dish was illuminated from below by an infrared (IR) light-emitting diode (LED) (not shown in Fig. 1c). A square black-and-white grating with a spatial period of 10 mm was projected onto the screen by a commercial digital light processing (DLP) projector (ASUS, Taiwan). Larvae were imaged through a macro objective (Navitar, USA) and an IR-pass filter with an IR-sensitive camera (Pike, Allied Vision Technology, Germany, or XIMEA, Germany) at 200 frames per second. The functional imaging rig was built in a similar way, with the two differences:

1. IR LED illuminating the screen with the chamber was directed from above (not shown in Figs. 3a and 6a) and the image was reflected on a hot mirror to reach a camera (XIMEA, Germany).
2. DLP projector used to provide visual stimulation (Optoma, USA) was mounted with a red-pass filter to avoid bleed-through of the green component of the visual stimulus in the light collection optics.

Stimulus presentation and tail tracking were controlled by the open-source, integrated system for stimulation, tracking and closed-loop behavioral experiments (Stytra)[68]. Larvae were presented with the grating moving in a caudal to rostral direction at 10 mm/s. Experiments were performed in closed-loop (similar to ref. [21]), as described below. Before starting an experiment, two anchor points enclosing the tail were manually selected. The tail between the anchor points was automatically divided into eight equal segments, and the angle of each segment with respect to the longitudinal reference line was measured by Stytra in real time.

The cumulative sum of the segment angles (measured in radians) constituted the final tail trace (top green trace in Fig. 1d). Sliding standard deviation of the tail trace with a time window of 50 ms was computed in real time, referred to as vigor. Vigor is a parameter that is close to zero when larvae do not move their tail and increases when they do, and can be therefore used to estimate the forward velocity that a head-restrained larva would have reached if it was freely swimming. To compute estimated velocity, the vigor was multiplied by a factor that was optimized during the initial calibration phase of each experiment (see "Experimental protocols" section) so that that the median estimated velocity during a typical bout was 20 mm/s (bottom green trace in Fig. 1d), which corresponds to a freely swimming condition. Swimming bouts were automatically detected in real time by comparing current estimated fish velocity with a set threshold of 2 mm/s. To provide behaving larvae with visual reafference, the estimated velocity was subtracted from the initial grating velocity during detected bouts (bottom magenta trace in Fig. 1d). As a result, larvae could experience the sensory consequences of their own swimming, despite being head-restrained. The initial and actual presented grating velocities, tail trace, vigor, estimated fish velocity, reafference condition (see below), and a binary variable denoting whether the fish was performing a bout or not at each acquisition frame constituted the raw data saved after each experiment (Fig. 1d).

Importantly, such closed-loop assay enables the experimenters to control and manipulate the reafference that animals receive when they swim and to study how perturbations in reafference affect behavior. The reafference perturbations used in this study can be grouped into three distinct categories (Fig. 2a):

1. The first type of perturbation, which has been previously used in the literature[21,22], is called *gain change*. In the closed-loop experimental assay, the gain parameter was used as a multiplier that converts the estimated swimming velocity of the larva into presented reafference. Therefore, the actual forward velocity of a swimming larva was proportional to gain. If the gain was set to zero, the tail movements had no influence over the grating speed, so larvae did not receive any reafference. This reafference condition was therefore referred to as the open-loop condition. If the gain was 1, the median velocity of the larva during a typical bout was 20 mm/s, referred to as the normal reafference condition. The gain values used in the experiments included 0, 0.33, 0.66, 1, 1.33, 1.66, and 2.

2. The second type of perturbation is called *lag*, and this corresponds to delaying the reafference with respect to the bout onset. When the lag was greater than zero, normal visual reafference with gain 1 was presented with a certain delay after the bout had started. The lag values used in the experiments included 0 ms lag (corresponds to the normal reafference condition), 75, 150, 225, 300 ms, and infinite lag (i.e., reafference never arrives after the bout onset, which is equivalent to the open-loop condition). In the shunted lag version of this condition, the reafference was set to 0 upon termination of the bout.

3. The third type of perturbation was called *gain drop*, and this corresponds to dividing the first 300 ms of a bout into four 75 ms segments and setting the gain during one or more of these segments to zero. Gain drop conditions were labeled using four-digit barcodes, where each digit represents the gain during a corresponding bout segment. For example, the gain profile 1100 denotes that the gain during bout segments 3 and 4 (i.e., from 150 to 300 ms after the bout onset) was set to zero, and during the rest of the bout, it was set to one. Gain drop conditions used in the experiment included 1111 (normal reafference condition), 0111, 0011, 0001, 0000, 1110, 1100, and 1000.

No combinations of reafference conditions were used, e.g., if the gain was different from 1, the lag was automatically set to 0 ms, or if the lag was greater than 0 ms, the gain was set to 1, and in both cases the gain drop was set to the normal 1111.

Note that reafference conditions listed above and presented in Fig. 2a are redundant. For example, the gain drop profile 0011 is exactly the same as 150 ms shunted lag, or gain 0 is exactly the same is infinite lag. Reafference conditions are presented in a redundant way to highlight that infinite lag makes a logical sense at the end of the list of lag conditions, and gain 0 makes a logical sense in the beginning of the gains list. The exact non-redundant list of all reafference conditions (18 conditions in total) is presented below:

- normal reafference (gain 1, 0 ms lag, and gain drop 1111);
- open-loop (gain 0 or infinite lag);
- gains: 0.33, 0.66, 1.33, 1.66, 2 (0 ms lag and gain drop 1111);
- lags and shunted lags: 75, 150, 225, and 300 ms (gain 1);
- gain drops: 1110, 1100, 1000 (0 ms lag).

## Experimental protocols

*General structure of experimental protocols.* All experimental protocols used in this study had a similar general structure. Each protocol consisted of trials. Each protocol consisted of a 15 s presentation of the grating moving in a caudal to rostral direction at 10 mm/s, preceded and followed by 7.5 s periods of the static grating

(30 s in total, see top magenta trace in Fig. 1d). Trials were grouped into four phases, unless otherwise specified (Fig. 4a):

1. Calibration phase (trials 1:10). During this phase, the multiplier defining how vigor is converted into estimated fish velocity was automatically calibrated so that the median velocity during an average swimming bout was 20 mm/s. Reafference condition during this phase was set to normal. This calibration was implemented to equalize velocity estimation across fish. In addition, during this phase, larvae were able to get used to the experimental environment and bring their swimming behavior to a stable level. All parameters recorded during this phase were not analyzed in this study.

2. Pre-adaptation phase (trials 11:20). During this phase, reafference condition was set to normal. This phase was used to record the baseline level of behavior, before any perturbations in reafference were introduced.

3. Adaptation phase (trial numbers depend on particular experimental protocol, see below). During this phase, larvae experienced perturbations in visual reafference.

4. Post-adaptation phase (last ten trials unless otherwise specified). During this phase, reafference condition was again set to normal. This phase was introduced to measure how the adaptation phase affected the baseline behavior.

*Acute reaction protocol.* This experiment was designed to probe the space of acute reactions to different unexpected perturbations in reafference and to test whether a feedback control mechanism can implement these reactions. The adaptation phase of the protocol consisted of 210 trials, and reafference condition for each bout performed during this phase was randomly selected from the list of 18 possible reafference conditions (see above; see also Fig. 2a). Bouts performed during other experimental phases were not included in the analysis.

*Whole-brain functional imaging protocol.* The protocol started with 2 min of no visual stimulation (projector presenting a black square) to record spontaneous activity. This was followed by a calibration phase, a pre-adaptation phase and an adaptation phase (40 trials). Post-adaptation phase was omitted. In four out of six imaged larvae, the calibration phase was omitted because behavior under the light-sheet setup was less consistent than under purely behavioral rigs, and calibration of the swimming velocity often failed due to insufficient number of bouts performed during this phase. For these larvae, the multiplier defining how vigor converts into estimated swimming velocity was set manually to a value resulting from successful calibration in one of the other two larvae. During the adaptation phase, reafference condition for each bout was randomly set to either normal or open-loop. In addition, two 350 ms pulses of reverse grating motion (rostral to caudal direction) at 10 mm/s were presented during each static grating period (5 and 10 s after the grating stopped moving). Responses to these pulses were not analyzed in this study. Furthermore, the difference in bout-triggered responses between normal reafference and open-loop condition was also not analyzed.

*Long-term adaptation protocol.* This experiment was designed to test if larval zebrafish can adapt their behavior in response to a long-lasting and consistent perturbation in reafference. The adaptation phase of the protocol consisted of 210 trials, and reafference condition for all bouts performed during this phase was set to 225 ms lag (Fig. 4a). This reafference condition was chosen because it elicited robust acute reaction (Fig. 2c_{ii}). In normal-reafference control larvae, reafference condition during the whole experiment (including the adaptation phase) was set to normal.

*PC functional imaging protocol.* Experimental protocol used for PC functional imaging experiment was a modified version of the long-term adaptation experiment (Fig. 6b). The adaptation phase was shortened from 210 trials to 50 trials to ensure stable recordings from PCs throughout the whole experiment. Such shortening was possible because long-term adaptation effects could already be observed after 50 trials of adaptation (Fig. 4c). On the other hand, the post-adaptation phase was prolonged from 10 trials to 50 trials to allow larvae to bring their behavior back to the original level after the adaptation.

**Behavioral data analysis**. Analysis of the behavioral data was performed in MATLAB (MathWorks, USA).

Recorded tail traces were z-scored and interpolated together with the grating speed traces to a common time array with a sampling period of 5 ms. For each swimming bout automatically detected by Stytra during the experiment, individual tail flicks were detected. One tail flick was defined as a section of the tail trace between two adjacent local extrema, with the magnitude greater than 0.14 rad and the duration not greater than 100 ms. Automatically detected onsets and offsets of the bouts were then corrected to coincide in time with the beginning of the first tail flick and the end of the last flick, respectively.

Only bouts that occurred while the grating was moving forward were considered for further analysis. For each bout, its duration and duration of the subsequent interbout was measured. If a bout was the last in a trial, the

corresponding interbout duration was not computed. All bouts that were shorter than 100 ms or had a subsequent or preceding interbout shorter than 100 ms were excluded from the analysis as potential tail tracking artifacts.

If there was at least one block of ten consecutive trials without any bouts, this animal was excluded from analysis. This was done because lack of reliable optomotor response might have indicated some damage caused during handling or the embedding procedure, or some severe behavioral or sensory deficits. The final numbers of included animals are listed below:

- Acute reaction experiments: 100 *TL* larvae (wild-type experimental group), 28 *Tg(PC:epNtr-tagRFP)*$^{-/-}$ larvae (treatment control group), and 39 *Tg(PC:epNtr-tagRFP)*$^{+/-}$ larvae (PC ablation group);
- Long-term adaptation experiments:

    ○ normal-reafference control animals: 103 *TL* larvae, 85 *Tg(PC:epNtr-tagRFP)*$^{-/-}$ larvae, and 83 *Tg(PC:epNtr-tagRFP)*$^{+/-}$ larvae,
    ○ lag-trained animals: 100 *TL* larvae, 85 *Tg(PC:epNtr-tagRFP)*$^{-/-}$ larvae, and 90 *Tg(PC:epNtr-tagRFP)*$^{+/-}$ larvae.

To analyze the temporal dynamics of the tail beat amplitude within individual bouts, a parameter termed bout power was computed as described below. A 1.1-second-long section of the tail trace (cumulative sum of eight tail segment angles measured in radians, see Closed-loop experimental assay in head-restrained zebrafish larvae in Methods above) was selected for each bout, starting from 100 ms before the onset of that bout. The values of the tail trace after the bout offset were replaced with zeros to exclude subsequent bouts that could occur within this time window. In addition, the median baseline value computed for the 100 ms window before the bout onset was subtracted from the section. Resulting sections of the tail trace were then squared and referred to as bout power, measured in squared radians.

To present the results of acute reaction experiments, we averaged the metrics obtained for each bout (its duration, subsequent interbout duration and power profiles) across bouts within each reafference condition in each larva. Bout and interbout duration are presented as mean ± SEM across larvae. Bout power profiles are presented as median across larvae.

To identify the time points, at which mean bout power depended on reafference condition, we used Kruskal–Wallis test (significance level 5%; Bonferroni correction for the total number of 220 tested time points). According to the test results, the bout power curves were then divided into ballistic and reactive periods (from 0 to 220 ms after the bout onset and from 220 ms onward, respectively) and the areas below the curves within these two periods were measured for each bout for each larva.

To present the results of the long-term adaptation experiments, we analyzed the aforementioned parameters only for the first bout in each trial. First bout duration in each trial is presented as mean ± SEM across larvae. To quantify the effects observed in the long-term adaptation experiments (acute reaction, reduction of acute reaction and the after-effect), we divided all trials of the protocol into blocks of ten and computed the mean value of respective parameter within each block. We then computed the differences between two corresponding blocks: acute reaction: first ten adaptation trials minus pre-adaptation trials, reduction of acute reaction: last ten adaptation trails minus first ten adaptation trials, after-effect: post-adaptation trials minus pre-adaptation trials. These quantifications are presented as median and interquartile range across larvae. To determine statistical significance of the observed differences between experimental groups, we used Mann–Whitney *U*-test with a two-tailed alternative with significance level of 5%.

In the long-term adaptation experiments performed under the light-sheet microscope, the lag-trained animals were sub-divided into adapting and non-adapting based on the reduction of acute reaction. Therefore, the more correct label for this group would be "not reacting to changes in visual feedback". We use the term "non-adapting" for convenience. If the first bout duration averaged across the last ten trials of the adaptation phase was less than that for the first ten trials of the adaptation phase by at least 40 ms, this lag-trained larva was considered adapting. To determine statistical significance of the long-term adaptation effects in lag-trained adapting larvae, we used Mann–Whitney *U*-test with a one-tailed alternative (with significance level of 5%) because the alternative hypothesis was already known from the main long-term adaptation experiment.

**Feedback control model of acute reaction**. To test whether acute reaction can be explained by a simple feedback controller that does not involve computation of expected sensory reafference (i.e., forward internal models), we developed a model that does not perform these computations (Fig. 2b) and tested its ability to adapt its output to perturbations in reafference (Fig. 2c).

The model was developed and tested in MATLAB (MathWorks, USA). The input of the model was the current velocity of the moving grating, and the output was a binary variable representing swimming velocity of the model. For simplicity, we did not set out to model individual tail flicks and approximated the swimming behavior of the zebrafish larvae by a binary motor output that equaled 20 mm/s when the model was swimming and 0 otherwise. This was possible due to the discrete nature of zebrafish swimming behavior at larval stage. Since in this study

we mainly focused on duration of bouts and interbouts, this simplification did not limit the ability to compare the model behavior with behavior of the real larvae.

To design the model, we used the results of the acute reaction experiment as a starting point (Fig. 2c–e). Thus, since larval zebrafish reacted to changes in visual stimulus with a fixed delay of 220 ms (Fig. 2e), the input of the model at a given time point was the grating velocity 220 ms before that point. We then assumed that forward motion of the grating should have a positive influence on the motor output, whereas reverse motion should have a negative influence. This assumption was based on the fact that if the grating was moving forward during a bout (as under open-loop or low gain conditions), this bout was significantly longer than under normal reafference condition, when the grating was moving backwards (compare cases for high and low gains in Fig. 2c$_i$). To implement this notion in the model, the input signal was split into forward and backward components by positive and negative rectification. This was performed by two respective nodes of the model: forward and reverse velocity sensors. Rectified signals were then recombined together with independent positive and negative weights ($\omega_f$ and $\omega_r$, respectively). We then proceeded from the fact that when a larval zebrafish was presented with a forward moving grating, it performed a swimming bout only after a certain latency period (for example, see Figs. 1d and 4b), suggesting that it integrates sensory evidence of the forward grating motion in time until the level of integrated signal reaches a motor command threshold. We therefore introduced a leaky velocity integrator (VI) that integrated recombined output of the velocity sensors with a time constant $\tau_s$. Output of the VI can be interpreted as motivation or sensory drive to swim because it increases with longer or faster forward motion of the stimulus, decreases with backwards motion, and drives activity in the motor part of the controller (see below). Activity of the VI was not allowed to be less than 0 (no sensory drive) or greater than 1 (maximum sensory drive). The sensory drive was then fed forward to the motor output generator (MOG), and this process can be interpreted as translation of the sensory input signal into motor coordinates. Accordingly, activity of the MOG was called motor drive. Whenever the motor drive reached a threshold thr, it activated the motor command generator (MCG), and the output of the model was set to 20 mm/s instead of 0. To ensure that the swimming bouts performed by the model do not last forever, we introduced a leaky motor integrator (MI) that integrated the output of the MCG in time with an input weight $\omega_m$ and a time constant $\tau_m$. Output of the MI can be interpreted as a metric of "tiredness" that encapsulates possible reasons for bout termination even when high sensory drive incites to continue swimming. This is the case, for example, under open-loop reafference condition, when the grating is moving forward and the sensory drive accumulates even though fish swims and tries to reduce the sensory drive. To implement the inhibitory influence of this "tiredness" on the motor output of the model, the MI inhibited the MOG with a weight $\omega_i$, thus reproducing a self-evident fact that the longer a bout had been so far, the sooner it would stop. In simple words, if the model was "tired" it had less motor drive to continue swimming, even if the sensory drive was strong. Activity of the MI was not allowed to be greater than 1 (maximum "tiredness"), and it could not inhibit the MOG below 0 (motor drive could not be negative). Finally, to ensure that bouts can last for some time once started in cases when the sensory drive to continue swimming becomes too low immediately after bout onset (for example, under high gain conditions when larvae receive a lot of reafference), we introduced a self-excitation loop to the MCG with a weight $\omega_s$ (see Supplementary Fig. 1a for formal mathematical description of the model).

Therefore, in total, the model had eight parameters. The input and output of the model, as well as activity of its nodes in an example trial with normal reafference are presented in Supplementary Fig. 1a.

To evaluate the ability of the model to acutely react to different perturbations in reafference, it was tested in a shorter version of the acute reaction experimental protocol. The protocol was shortened to save the computation time required for fitting the model. One trial consisted of 300 ms of static grating followed by 9.7 s of the grating moving in a caudal to rostral direction at 10 mm/s. The reafference condition of the first bout was always normal, and the reafference condition of the second bout was chosen from a list of 18 reafference conditions used in the acute adaptation experiment (see Closed-loop experimental assay in head-restrained zebrafish larvae in Methods above). If the model initiated a third bout, the trial was terminated, and the duration of the second bout and subsequent interbout constituted the final output of the model in that trial. If the model did not initiate the third bout, the final output of the model in that trial was not computed.

The parameters of the model were fitted to training datasets obtained for each larva that participated in the acute reaction experiment ($N = 100$) using a custom-written genetic algorithm. To obtain the training datasets, 18 arrays of bout durations and 18 arrays of interbout durations were generated for each larva, each array corresponding to one reafference condition. Average values of randomly selected 50% from each array constituted the training datasets, the remaining 50% were used as test datasets. Distributions of resulting parameter solutions are presented in Supplementary Fig. 1b.

The optimization algorithm minimized the mean absolute error (absolute difference between the output array of the model and a training dataset, normalized by the training dataset) and resulted in sets of the model parameters, each optimized to fit one larva. To present the results, mean ± SEM of the final output arrays of models across all sets of parameters, and of the test datasets across all larvae were computed (Fig. 2c, d).

**Light-sheet microscopy**. Functional imaging experiments were employed in this study for two purposes: to test the main assumption of the feedback control model (the existence of the sensory integration in the larval zebrafish brain) and to test whether activity of PCs can represent the output of an internal model. Respectively, there were two types of functional imaging experiments: whole-brain imaging experiments and PC imaging experiments. Both were performed using a custom-built light-sheet microscope (Figs. 3a and 6a).

For the whole-brain imaging (Fig. 3a), a beam coming from a 473 nm laser source (modulated laser diodes, Cobolt, Sweden) was split with a dichroic mirror and conveyed to two orthogonal scanning arms. Each scanning arm consisted of a pair of galvanometric mirrors (Sigmann Electronik, Germany) that allowed vertical and horizontal scanning of the beam, a line diffuser (Edmund Optics, USA), a scan lens (Thorlabs, USA), a paper screen to protect the fish eyes from the laser, a tube lens (Thorlabs, USA), and a low numerical aperture objective (Olympus, Japan). The emitted fluorescence was collected through a water immersion objective (Olympus, Japan) mounted on a piezo (Piezosystem Jena, Germany), band-pass filtered (AHF Analysentechnik, Germany), and focused on a camera (Orca Flash v4.0, Hamamatsu Photonics K.K., Japan) with a tube lens (Thorlabs, USA).

The piezo, galvanometric mirrors, and camera triggering were controlled by a custom-written Python program. The light-sheets were created by horizontal scanning of the laser beams at 800 Hz. The light-sheets and the collection objective were constantly oscillating along the vertical axis with a saw tooth profile of frequency 1.5 Hz and amplitude of 250 μm. At each oscillation, 35 camera frames were acquired at equally timed intervals, with an exposure time of 5 ms. The resulting volumetric videos had a voxel size of $0.6 \times 0.6 \times 7$ μm, and a sampling rate of 1.5 Hz per volume.

For the PC imaging experiments, the frontal scanning arm was removed because the whole cerebellum could be illuminated using only the lateral beam (Fig. 6a). The sampling rate was increased to 4 or 5 Hz due to smaller volume of imaging.

Larvae were continuously illuminated with the excitation blue light throughout the entire imaging sessions, which could potentially interfere with the behavior. However, behavior performed under the light-sheet microscope was overall comparable with the behavior performed in the behavioral rigs. Thus, for example, normal reafference control larvae performed, on average, $12.1 \pm 0.4$ bouts per trial in the behavioral rigs and $13.5 \pm 3.0$ under the light-sheet microscope (mean ± SEM across larvae), and the shape of the bouts was similar (compare Figs. 3c, 6di and Supplementary Fig. 6a with Figs. 1d, 4b). Furthermore, acute reaction and long-term adaptation effects were also present under the light-sheet and comparable with the behavioral rigs (compare Fig. 6c and Supplementary Fig. 5 with Fig. 4c–f).

**Functional imaging data analysis**. Behavioral data acquired during the imaging experiment were analyzed as described in "Behavioral data analysis" section. This section describes analysis of the imaging data.

*Whole-brain functional imaging data analysis*. To analyze the functional imaging data, we first preprocessed them in Python following the methods presented in ref. [32]. To correct the data for motion artifacts and potential drift, they were aligned to an anatomical reference generated by averaging the first 1000 frames of each plane. For each volume in time, the translation with respect to the reference volume was computed by cross-correlation using the *register_translation* function from the *scikit-image* Python package[67]. Before the alignment, the reference volume and volumes to be registered were passed through a Sobel filter after a Gaussian blur (with the standard deviation of 3.3 voxels) to emphasize image edges over absolute pixel intensity. Volumes for which the computed shift was larger than 15 voxels (generally due to large motion artefacts caused by vigorous tail movements of the embedded fish) were discarded and replaced with NaN values. For subsequent registration of the imaging data to a common reference brain (see below), a new anatomical stack was computed for each animal by averaging the first 1000 frames of the aligned planes.

To segment the imaged volume into regions of interest (ROIs), a "correlation map" was computed, where each voxel value corresponded to the correlation between the fluorescence time-trace of that voxel and the average trace of eight adjacent voxels in the same plane. Then, based on the correlation map, individual ROIs were segmented in each plane with the following iterative procedure. Growing of each ROI was initiated from the voxel with the highest intensity in the correlation map among the ones still unassigned to ROIs, with a minimum correlation of 0.3 (seed). Adjacent voxels were then gradually added to the growing ROI if eligible for inclusion. To be included, adjacent voxels' correlation with the average fluorescence time-traces of all voxels assigned to the ROI up to that point had to exceed a set threshold. The threshold for inclusion was 0.3 for the first iteration and increased linearly as a function of distance to the seed, up to a value of 0.35 at 3 μm distance. Additional criteria for minimal and maximal ROI area (9–28 μm$^2$) ensured that the ROIs matched approximately the size of neuron somata. After segmentation, the fluorescence time-trace of each ROI was extracted by summing fluorescence of all voxels that were assigned to that ROI during segmentation.

Subsequent analysis steps were performed in MATLAB (MathWorks, USA). To de-noise the traces, a low-pass Butterworth filter with a cutoff frequency of 0.56 Hz was applied to each trace. This frequency corresponds to the half-decay time of the calcium indicator GCaMP6s expressed by the imaged larvae (1.8 s[69]), and

fluorescence oscillations at frequency higher than that were unlikely to result from biological events. In addition, to correct for potential slow drift, the drifting baseline of each trace was computed by applying a low-pass Butterworth filter with a cutoff frequency of 3.3 mHz; and this baseline was then subtracted from the trace. The traces were then *z*-scored for subsequent analysis.

The strategy of subsequent analysis was aimed to identify velocity integrators within the larval zebrafish brain and included two steps.

The first step was aimed to identify ROIs that responded to the forward moving grating. Since this stimulus reliably triggers swimming behavior, it was important to disambiguate ROIs with responses to the forward moving grating from ROIs with motor-related activity. To this end, we computed the average grating- and bout-triggered fluorescence for each trace. To do this, we selected 5 s long sections of the trace, starting from one second before the respective trigger. To avoid contamination of triggered responses by activity resulting from other events, we only considered triggers that did not have any other triggers in the preceding second, and if another trigger occurred within the selected time window, all fluorescence values after this another trigger were replaced with NaN values. In addition, we subtracted the baseline, defined as mean fluorescence before the trigger within the selected time window, from each triggered trace. We then computed the average traces and SEM across corresponding triggers for each trace. To identify ROIs with sensory-related and motor-related activity (referred hereafter as sensory and motor ROIs), we first computed the mean values of average grating-triggered fluorescence within the time window from 0 to 4 s after the grating onset, and of average bout-triggered fluorescence within the time window from 0 to 2 s after the bout onsets. Obtained values were referred to as sensory and motor scores. To estimate the probability of observed scores to result from chance, we then divided each trace into 84 sections, each 23 second-long, randomly shuffled the sections 1000 times and computed sensory and motor scores for each shuffling to build null-distributions of the scores. If the actual score was greater than 95th percentile of the respective null-distribution, it was considered significant. ROIs were defined as sensory if they had a significant sensory score. If in turn, an ROI had a significant motor score and a non-significant sensory score, it was defined as a motor ROI. This additional criterion for definition of a motor ROI was introduced because almost all sensory ROIs continued to increase their grating-triggered fluorescence during swimming bouts. As a result, many sensory ROIs had a significant motor score, so this parameter alone could not be used to define motor ROIs.

The second step was aimed to identify whether some of the sensory ROIs integrate sensory evidence of the forward moving grating in time. To this end, we fitted a leaky integrator model to the fluorescence traces of sensory ROIs by iterating over a range of time constants from 0.5 to 10 s, with 0.5 s steps (sampling period duration), and identifying the time constant resulting in the highest correlation between the model and the actual trace. The leaky integrator trace was additionally convolved with a GCaMP6s kernel, modeled as an exponential function with a half-decay time of 1.8 s[69]. Note that testing time constants shorter than 0.5 s was not possible due to relatively low sampling rate (2 Hz). Therefore, if a given ROIs had an estimated time constant of 0.5 s, the true value of the time constants lies between 0 and 0.5 s. ROIs with short time constants (≤1.5 s) were referred as sensors, whereas ROIs with longer time constants were called integrators.

To compare the location of ROIs assigned to the aforementioned functional groups across larvae and to present the ROIs in the context of gross larval zebrafish neuroanatomy, the imaging data was registered to a common reference brain using the free Computational Morphometry Toolkit[70]. To this end, affine volume transformations were computed to align the anatomical stacks from each larva to the reference brain. Computed transformations were then applied to each ROI to identify its coordinates in the reference space. To present the final ROI maps, binary stacks with ROIs of a given functional group were summed across larvae, and the maximum projections along dorsoventral and lateral axis were computed. In addition, to identify the anatomical regions with experiment-related activity, the regions annotated in the Z-Brain atlas[71] were registered to our reference brain using the same procedure.

If an ROI was assigned to one of the three aforementioned functional groups (sensors, integrators, or motor ROIs), it was referred to as active ROI. To determine whether anatomical location of active ROIs was consistent across larvae, we first formulated a null-hypothesis for each ROI. The hypothesis stated that active ROIs that spatially overlap with this ROI in a population of larvae (i.e., ROIs that occupy the very same anatomical region in the brain) are equally likely to be either sensors, or integrators, or motor ROIs. According to this null-hypothesis, the probability of a given active ROI to be assigned to any functional group is 1/3. We then tested this null-hypothesis for each ROI against the one-tailed alternative that a given ROI was more likely to be assigned to its actual functional group than to the other two groups with a significance level of 5%. Rejection of the null-hypothesis can be interpreted as that in the brain region corresponding to this ROI, the probability of finding active ROIs of the same functional group in a population of larvae is greater than that of finding active ROIs of the other two groups. To test the null-hypothesis, we first calculated the number of larvae that had any active ROIs overlapping with the original ROI and the number of larvae that had an overlapping ROI assigned to the same functional group as the original ROI. Then, the probability of this observation given the null-hypothesis was inferred using

maximum likelihood estimation. If this probability was less than 5%, the null-hypothesis was rejected and the anatomical region corresponding to the original ROI was concluded to be consistent across larvae.

*PC functional imaging data analysis.* PC functional imaging data was pre-processed in Python. Before entering data in the analysis pipeline, data was previewed blindly with respect to experimental condition and behavioral performance. Data that showed any sign of drifting during the whole duration of the experiment were discarded, to avoid any confounding effect in the subsequent analysis. After this selection, $N = 25$ larvae were kept out of the original 50 ($N = 8$ normal-reafference control larvae and $N = 17$ lag-trained larvae). Lag-trained larvae were further subdivided into adapting ($N = 9$) and non-adapting ($N = 8$) as described in "Behavioral data analysis" section.

Compared with the whole-brain data, cerebellum imaging data were smaller and PC labeling was sparser, so this dataset was better suited for signal extraction using Suite2p[72]. Suite2p was used for plane-wise alignment of the data and ROIs segmentation; after these steps, the raw extracted fluorescence was used in subsequent analyses, bypassing the spike deconvolution part of the Suite2p pipeline. Parameters used for the extraction were the Suite2p default values for 2p detection, except for expected cell size and temporal resolution that were adjusted according to the imaging settings. Manual curation was performed on each fish, blindly with respect to experimental group and behavioral performance, to exclude artifactual ROIs segmented from the skin visible in the imaging.

Subsequent analysis steps were performed in MATLAB (MathWorks, USA). We first modeled a motor regressor by convolving the binary variable representing whether the fish was performing a bout or not at each acquisition frame with a GCaMP6s kernel, modeled as an exponential function with a half-decay time of 1.8 s[69] (Supplementary Fig. 6a). This regressor was processed in the same way as fluorescence traces of real ROIs, as described below.

To correct for potential slow drift, each trace was passed through a high-pass Butterworth filter with a cutoff frequency of 3.3 mHz. The traces were then z-scored for subsequent analysis.

Subsequent analysis steps are illustrated in Fig. 6d. We first computed the first-bout-triggered fluorescence for each trace in each trial by selecting 2.2 s-long sections of the trace, starting from 1 s before the first bout. The baseline, defined as mean fluorescence before the first bout onset within the selected time window, was subtracted from each triggered trace. We then computed first-bout-triggered responses by averaging the triggered traces within the time window from 0 to 1.2 s after the first bout onsets. These responses were then averaged within the block of ten trials, similarly to the first bout duration (see "Behavioral data analysis" section).

We used these averaged responses to define the four criteria for each trace, which represented how much the responses have changed during important transitions of the experimental protocol.

1. Criterion 1 was computed as the difference between the first ten trials of the adaptation phase and ten trials of the pre-adaptation phase.
2. Criterion 2 was computed as the difference between the last ten trials and the first ten trials of the adaptation phase.
3. Criterion 3 was computed as the difference between the first ten trials of the post-adaptation phase and the last ten trials of the adaptation phase.
4. Criterion 4 was computed as the difference between the last ten trials and the first ten trials of the post-adaptation phase.

Obtained criteria were converted into 4-digit ternary barcodes, each digit corresponding to one criterion and can take values "+", "0", or "−", using the following bootstrapping procedure. For each trace we computed a null-distribution of criteria under the assumption that observed changes in bout-triggered responses were not related to transitions in the experimental protocol. Thus, to build the null-distributions, the trials (excluding the calibration phase) were randomly shuffled 100,000 times, and criterion 1 was computed in each shuffling repetition. If a given criterion was greater than the 97.5th percentile of the null-distribution for that ROI, the corresponding criterion was replaced with "+", if it was less than the 2.5th percentile, it was replaced with "−", and otherwise with "0". This allowed us to assign 4-digit barcodes to each ROI, and all ROIs were categorized into clusters based on these barcodes. We have verified that the key finding of this experiment (namely that the only cluster significantly enriched in the lag-trained adapting group was 0-0+) was not sensitive to which exact percentiles were used during bootstrapping.

To identify activity profiles with different occurrences across experimental groups, we computed the fractions of ROIs assigned to each cluster within each larva. Statistical significance of observed differences in fractions was determined using Kruskal–Wallis test (significance level 5%).

To compare the location of ROIs across larvae, the imaging data was registered to a common reference cerebellum similarly to the whole-brain imaging data (see above). To present the final maps of 0-0+ ROIs, we first applied 3D Gaussian blur with standard deviation of 1 μm to the binary stacks with these ROIs. The stacks were then summed across larvae, and the maximum projections along the dorsoventral axis were computed. 0-0+ ROIs were detected more frequently in the medial cerebellum (Supplementary Fig. 7—top row). To estimate whether such spatial segregation could have resulted from chance, we randomly sampled the same number of ROIs in each larva, computed the maps as described above, repeated this 100 times, and computed the final maps by averaging across 100 iterations.

**Reporting summary**. Further information on research design is available in the Nature Research Reporting Summary linked to this article.

## Data availability
All data generated in this study have been deposited in the zenodo[73]. Source data are provided with this paper.

## Code availability
All Python and MATLAB code used to acquire and analyze data used in current study is available in the zenodo repository[74].

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

## Acknowledgements

We thank Tugce Yildizoglu and Patricia Cooney for help with preliminary experiments. We thank Hagar Lavian for help with running experiments and Vilim Štih for discussions on feedback controllers. R.P. would like to thank Winfried Denk for multiple in-depth discussions concerning this project. All authors were funded by the Max Planck Society. D.A.M. and A.M.K. were funded by IMPRS for Life Sciences. A.M.K. was funded by a Joachim-Herz Foundation fellowship. L.P. was funded by GSN. D.A.M., L.P., and R.P. were funded by the Deutsche Forschungsgemeinschaft (DFG, German Research Foundation) through grant PO 2105/2–1. This work was funded by the Deutsche Forschungsgemeinschaft (DFG, German Research Foundation) under Germany's Excellence Strategy within the framework of the Munich Cluster for Systems Neurology (EXC 2145 SyNergy—ID 390857198).

## Author contributions

D.A.M., and R.P. conceived of the project. D.A.M. performed most of the experiments, analyzed the data, developed the model and created the figures with help from L.P. L.P. performed functional imaging experiments and pre-processed the imaging data. A.M.K. generated the Tg(PC:epNtr-tagRFP) zebrafish line and developed the *gain drop* paradigm. D.A.M. and R.P. wrote the manuscript with help from all authors.

## Funding

## Competing interests

The authors declare no competing interests.
