## [Peer Review File · Nature Communications]

A cerebellar internal model calibrates a feedback controller involved in sensorimotor controlREVIEWER COMMENTS

Reviewer #1 (Remarks to the Author):

The manuscript by Markov and colleagues addresses adaptation of the optomotor response (OMR) in larval zebrafish, detailing both short- and long-timescale changes to swimming features in response to manipulations of visual feedback. The authors ultimately aimed to understand which aspects of swimming adaptation are mediated by the cerebellum. The initial experiments built on past studies of optomotor adaptation in zebrafish, including a rich complement of parametric perturbations to reafference magnitude and timing that elicited a host of effects on the duration and timecourse of swimming vigor. A thoughtfully devised model recapitulated the relationship between reafference and swimming duration, providing compelling evidence that a feedback controller is sufficient to account for the short-term modifications to swimming newly observed as well as those described previously. Further support was provided by the identification and localization of ROIs serving as stimulus integrators using calcium imaging. The authors also showed long-term changes to swimming that could not be mediated by feedback control, and demonstrated long-term effects on swim duration after lesioning Purkinje cells. Finally, the authors identified Purkinje cell ROIs with activity profiles consistent with the implementation of an internal model for adapting swim duration - gradually scaled activity under repeatedly delayed sensory feedback, and gradual recovery after sensory feedback was restored (to simulated normal timing). The story beautifully integrates feedback and internal model control of a single behavior. The addressed problem is important, the manuscript was clearly written, the introduction and discussion covered relevant concepts and background, the figures were clear and informative, and the experiments were carefully designed and rigorously implemented. Despite the manuscript's many strengths, its key conclusion - that the cerebellum implements an internal model for long-term changes to OMR - requires further analysis to be justified.

Major concerns

Purkinje cell lesion appears to prolong bout duration, complicating conclusions about the role of the cerebellum in long-term adaptation and implementation of an internal model. The results as presented do not rule out that the adaptation mechanism is intact in lesioned animals, reducing bout duration a fixed amount. The authors claim "our loss-of-function experiments have demonstrated that... acute reaction is a cerebellum-independent process (Fig. 5b-c; Extended Data Fig 4)" (L503). However, these figures appear to show the opposite. Lesioned animals exhibit a larger acute reaction (Figure 5c), as well as the appearance of generally prolonged bouts and shorter inter-bouts (Extended Fig 4). The statement on L346 could therefore be clearer: "PC ablation impairs only the long-term adaptation to a consistently changed reafference condition, while sparing the OMR itself..." Given the elevated acute reaction after lesion, the argument that lesion impairs long-term adaptation requires a direct contrast of the magnitude of adaptation (last 10 adaptation trials - first 10 adaptation trials) rather than the contrast shown in Figure 5d.

Statistics on activity changes across criteria 1-4 (Figure 6d) appear prone to false positives. Each contrast in each ROI in each animal had its significance assessed independently via bootstrapping with an alpha of 0.05 (L1358). This approach does not account for multiple comparisons including 4 criteria and numerous ROIs. Consistently, the preponderance of ROIs with significantly altered activity across criterion 1 in the control group (which experienced no change in stimulus across the relevant task phases) raises concern that spurious changes are being labeled “significant” with this approach (compare Figure 6ei first column control, non-adapting, and adapting). With that said, the significant difference in the number of “0-0+” cells in the adapting vs. control and non-adapting groups is compelling. It would be easier to interpret these data if the authors would list the number of cells in this category and total number of cells in each of the groups in the Results. Furthermore, the authors should indicate whether members of this barcode group remain if computing significance when accounting for multiple comparisons.

Minor concerns

Are the colors in Figure 2ei flipped? This panel conflicts with data in Figure 2ci and gain 0 in Fig 2ei doesn't match infinite lag in Fig 2eii.

The authors should speculate on which nodes in their control diagram might be under the influence of cerebellar/internal model output?

L215 - “metric of tiredness” explanation doesn't enhance understanding of the model. Relatedly, L443: “model suggests... duration is determined by the interplay between... sensory drive and tiredness.” Anthropomorphizing needs more explanation if the authors feel it justified.

A brief description of model parameter solutions would be informative. For instance, how do ω_f and ω_r compare? How do τ_v and τ_m compare?

I would expect “velocity integrator” ROIs to exhibit a slower return to baseline after grating stimulus offset, but Figure 3g appears to show similar decays for “sensor” and “integrator” ROIs.

The authors should show activity aligned to stimulus offset, similar to Figures 3h and 3i. If integrators do not relax more gradually, how does one reconcile differing onset and offset kinetics?

Figure 6c - Non-adapting animals don't appear to exhibit an acute response. In this case, a more accurate description would be “non-responding” than “non-adapting.” This is critical, because their response to the lagged reafference cannot be attenuated if it isn't existent in the first place. The authors

should show evidence, if existent, that the “non-adapting” group has an elevated bout duration during trials 21:30.

The authors should indicate in Experimental protocols when the light sheet was illuminated with respect to behavior during imaging sessions.

The control theoretic term “plant” deserves an introduction in line 41.

More detail would be helpful regarding computation of bout power. How was the “tail trace” derived, and what are its units? (L1059).

The lagged group purportedly became “statistically indistinguishable from the control group... (Fig 4e),” but that panel contrasts groups on the difference between adaptation and baseline levels, not on their durations directly.

It is unclear how the data in Extended Figure 6 evidence the figure’s title. Are the arrows and color coding meant to indicate lack of correlation between the behavior and activity? If so, the relationship should be shown more explicitly.

-signed

David Ehrlich

Reviewer #2 (Remarks to the Author):

General assessment and a summary of any substantive concerns

This manuscript by Markov and colleagues presents a study revealing how the zebrafish adjusts to visual feedback, thereby adjusting its own motor output in terms of frequency of swim bouts and their vigor. This study is a rigorous, well designed behavioral and imaging study that shows that zebrafish not only respond with predictable behavioral changes when this reafference is changed acutely but also that zebrafish learn over time to expect specific visual feedback. In summary, the authors make the claim that acute reaction to changed visual feedback can be implemented by a ‘simple’ feedback controller

that integrates the optic flow, which they claim to reside in specific brain areas (e.g. pretectum, thalamus) as evidenced by light-sheet functional calcium imaging. However, if visual feedback is altered using a lag in visual feedback for a long period of time, fish learn to slowly adapt which the authors claim is proof of a learned internal model. Purkinje cells (PC) seem to play a role in this learning as their cell type-specific chemogenetic ablation impairs long-term adaptation, but not acute adaptation. Thus, using acute and long-term manipulations of the visual feedback, the authors demonstrate that the neural circuitry underlying the OMR is and can be calibrated and needs to be recalibrated when visual feedback is changed abruptly.

While the experiments of acute motor adaptations in head-embedded zebrafish are not new, the rigor and detailed investigation of behavioral consequences in response to subtly altering the visual reafference, provide new insights on how visual feedback works in this context and a great data resource for the specific field. E.g. the finding and distinction that the initial 220 ms of a swim bout is stereotyped ballistic period is followed by a reactive period in which fish can affect the tail beat amplitude is relevant for modeling purposes. The authors provide a convincing feed forward controller mechanism that may not require learning, according to their modelling and discussion. The cerebellum has long thought to integrate sensory and motor-related information in Purkinje cells (PC), the main computational components that disseminate information to the motor systems and enable motor learning. In this case, the authors show that PCs mediate the long term adaptation by generating a new transgenic line that allows ablation of (most?) PCs in a well-controlled set of experiments. While it would be important to determine the mechanism, this might be more suited for a future detailed study.

The research appears rigorous, with clear figures, legends, well-executed and presented data are extensive and appropriately analyzed with relevant statistical methods, and the materials and methods section is extensive, well written, detailed, and clear.

Despite the many strengths of the current manuscript, some questions remain and may require revisions or discussion in the paper:

- 1) Given their demonstrated high technical abilities, why did the authors not perform whole-brain light-sheet recordings using the protocol of long-term adaptation? (Under non-COVID-19 circumstances,) this would be an easy and interesting experiment to investigate brain-scale learning and adaptation in question, potentially transforming the findings of this study as it both might reveal non-PC cerebellar activity or distributed activity that might be part of the 'internal model'. At the very least please discuss.
- 2) The authors should provide supplemental movies of exemplary behavior and neural recordings under various different conditions to allow the reader to a) better understand the manipulations and b) provide deeper experimental insight of the findings.

3) The authors should include single-cell examples (in supplement if necessary but better in the main figure) of functional imaging- for critical classes of ROI and how they compare to single neuron recordings. The omission of these kinds of examples might otherwise be interpreted that ROIs cannot represent single cells or important details cannot be resolved.

4) In Extended Data Fig. 1 a formal mathematical description of the controller model is provided, but it would be much preferred if the author would offer the associated Matlab code for all conditions and figures for reproducibility and rigor and future comparisons.

5) It would be nice to see a verification of the PC ablation, Extended Data Figure 3 appears ambiguous whether some PC neurons, especially dorsally remain intact. The image does not seem to provide conclusive evidence that all of them are gone. However, the effect is strong. Therefore, please provide quantification and possibly additional verification using higher resolution images that all PCs are gone, at least in an example fish. Discuss in the text to aid clarity of the findings.

6) Provided the strong evidence from behavioral data and neural recordings, the model in Figure 7 is relatively disappointing. Instead, no effort was made to integrate or quantitatively model the cerebellar/ PC role, compare and identify variables that account for the cerebellar internal model. Especially, PC specific model that would illustrate the predictions for perturbation experiments would be much preferred. For example, the authors could use the bout-triggered responses that gradually decrease during the adaptation period and increase back to the initial level when entering the post adaptation phase. A model that also would account for both proposed mechanisms should at least be attempted or discussed. One interesting point such a model could answer as well is that modeling realistic PC activation (even slow Calcium dynamics) might suggest a role for single PCs as well as for the population in terms of calibration/ learning.

7) In line 1162 and other places custom-written genetic algorithm. Please specify what that is. Like the point above, consider making the code available

8) Please make all data or at least exemplary data available for better reproducibility.

9) Please make the associated code available for better reproducibility.

10) Why is there no mechanistic discussion of overall PC circuit function?

11) Please expand discussion including relevant work in other species e.g. Kostadinov et al. 2019, Heffley, W. et al. 2018. But also, how these findings fit into recent work in zebrafish involving the cerebellum. Lin et al. 2020.

12) In the discussion, could the authors discuss how acute changes in afference blend with the internal model over time? The authors did not convincingly make the case that these two mechanisms are completely separate in intact animals, and it is expected according to their data that updating of the internal model begins after one bout.

Overall, the data appears to be of high quality, and the authors have gone to great lengths to establish extensive behavioral data sets for various incremental variations of visual reafference, provided large statistical samples of neurons, repetitions, and animals in each condition presenting a statistically sound

study throughout. Especially, the behavioral data provides excellent evidence for its conclusions that different neural mechanisms are at play. To the best of my knowledge, both the behavioral results, modeling, transgenic fish lines for PC ablation and imaging are all completely novel and this study represents both an important data resource for the specific field (e.g. answering how zebrafish respond to variations of acute and long-term visual feedback). More generally, the paper represents an advance in understanding whether and how internal model-based learning might occur or and will likely influence future studies on this topic.

If the above points would be addressed and a couple of cosmetic changes listed below would be made, the presented research is a good fit for Nature Communications. Both the insights into the visually guided behavior, visual reafference, and implication of cerebellar function in adaptation and technical execution are excellent, justifying publication in this journal.

Minor comments

Line 41: 'body plant' should be changed to 'body plan'

Line 62: 'body plant' should be changed to 'body plan'

Reviewer comments

We are very grateful to David Ehrlich and to an anonymous reviewer for their careful reading of this manuscript, for highlighting its importance and other strengths as well as for their insightful and valuable comments and suggestions. We have noted their contribution in the acknowledgements. The questions and requested revisions have been carefully considered and implemented to improve the manuscript.

Reviewer #1

The manuscript by Markov and colleagues addresses adaptation of the optomotor response (OMR) in larval zebrafish, detailing both short- and long-timescale changes to swimming features in response to manipulations of visual feedback. The authors ultimately aimed to understand which aspects of swimming adaptation are mediated by the cerebellum. The initial experiments built on past studies of optomotor adaptation in zebrafish, including a rich complement of parametric perturbations to reafference magnitude and timing that elicited a host of effects on the duration and timecourse of swimming vigor. A thoughtfully devised model recapitulated the relationship between reafference and swimming duration, providing compelling evidence that a feedback controller is sufficient to account for the short-term modifications to swimming newly observed as well as those described previously. Further support was provided by the identification and localization of ROIs serving as stimulus integrators using calcium imaging. The authors also showed long-term changes to swimming that could not be mediated by feedback control, and demonstrated long-term effects on swim duration after lesioning Purkinje cells. Finally, the authors identified Purkinje cell ROIs with activity profiles consistent with the implementation of an internal model for adapting swim duration - gradually scaled activity under repeatedly delayed sensory feedback, and gradual recovery after sensory feedback was restored (to simulated normal timing). The story beautifully integrates feedback and internal model control of a single behavior. The addressed problem is important, the manuscript was clearly written, the introduction and discussion covered relevant concepts and background, the figures were clear and informative, and the experiments were carefully designed and rigorously implemented. Despite the manuscript's many strengths, its key conclusion - that the cerebellum implements an internal model for long-term changes to OMR - requires further analysis to be justified.

Major concerns

Purkinje cell lesion appears to prolong bout duration, complicating conclusions about the role of the cerebellum in long-term adaptation and implementation of an internal model. The results as presented do not rule out that the adaptation mechanism is intact in lesioned animals, reducing bout duration a fixed amount. The authors claim "our loss-of-function experiments have demonstrated that... acute reaction is a cerebellum-independent process (Fig. 5b-c; Extended Data Fig 4)" (L503). However, these figures appear to show the opposite. Lesioned animals exhibit a larger acute reaction (Figure 5c), as well as the appearance of generally prolonged bouts and shorter inter-bouts (Extended Fig 4). The statement on L346 could therefore be clearer: "PC ablation impairs only the long-term adaptation to a consistently changed reafference condition, while sparing the OMR itself..." Given the elevated acute reaction after lesion, the argument that lesion impairs long-term adaptation requires a direct contrast of the magnitude of adaptation (last 10 adaptation trials - first 10 adaptation trials) rather than the contrast shown in Figure 5d.

It is true that PC ablation results in prolonged bouts and increased acute reaction to lagged feedback. Importantly, acute reaction is still present (and indeed increased) after PC ablation indicating that intact PCs are not required for such short-term motor adaptation. We agree that the term "cerebellum-independent" was misused; and it is now corrected in the revised text.

However, we do not think that this increase in acute reaction in PC-ablated animals can explain the observed impairment in long-term adaptation, particularly the after-effect. The after-effect shows that consistent long-term exposure to a modified feedback condition recalibrates the baseline behavior,

something that was not observed in PC-ablated group. In essence, if we define long-term motor adaptation as a process that leads to some changes in the behavior performed under exactly the same conditions as before adaptation, the after-effect is the most direct metric of the adaptation. The fact that the after-effect is absent in PC-ablated animals indicates that long-term adaptation is impaired by PC ablation.

Finally, to further support this reasoning, we have replaced the metric shown in Figure 5d to the metric suggested by Dr. Ehrlich: last 10 adaptation trials - first 10 adaptation trials. We agree that it reflects the long-term adaptation in a more direct way. Revised Figure 5d shows that in treatment control animals (as well as in wild-type controls, Figure 4e) the magnitude of acute reaction was decreased by the end of the adaptation compared to the beginning of the adaptation, and that this effect was **not** observed in PC-ablated fish. Therefore, the use of a suggested metric did not affect this conclusion.

Statistics on activity changes across criteria 1-4 (Figure 6d) appear prone to false positives. Each contrast in each ROI in each animal had its significance assessed independently via bootstrapping with an alpha of 0.05 (L1358). This approach does not account for multiple comparisons including 4 criteria and numerous ROIs.

Consistently, the preponderance of ROIs with significantly altered activity across criterion 1 in the control group (which experienced no change in stimulus across the relevant task phases) raises concern that spurious changes are being labeled “significant” with this approach (compare Figure 6e first column control, non-adapting, and adapting).

With that said, the significant difference in the number of “0-0+” cells in the adapting vs. control and non-adapting groups is compelling. It would be easier to interpret these data if the authors would list the number of cells in this category and total number of cells in each of the groups in the Results.

Furthermore, the authors should indicate whether members of this barcode group remain if computing significance when accounting for multiple comparisons.

The rationale behind assigning 4-digit barcodes to ROIs was to categorise ROIs into clusters, which allowed us to compare occurrences of different types of ROIs across experimental groups. An alternative approach could be to cluster ROIs based on actual values of criteria without converting them into ternary barcodes. However, this approach seemed dangerous because actual values of criteria might differ across ROIs for technical task-unrelated reasons. These reasons could include potentially different levels of GCaMP expression, different signal-to-noise ratio, different levels of excitation illumination and light scattering across ROIs and across fish, and, importantly, differences in sample drift during relatively long periods of recording over which the criteria were computed (20 trials for criteria 1 and 3, and 50 trials for criteria 2 and 4, which corresponds to 10 and 25 min of recording, respectively). As a result, many ROIs might end up clustered into one category because they all belong to one fish as opposed to representing a group of PCs with similar task-related properties. The advantage of bootstrapping is that it takes into account the statistics of a phenomenon under study when building null-distributions, and this is why it was used. Our goal was not to postulate that a given ROI has a statistically significant criterion in a true sense of “statistical significance” but rather to define some meaningful thresholds to convert the criteria into barcodes. The term “statistically significant” was therefore mis-used in the text, and the corresponding sections have been corrected.

Regarding the rate of false positives, we believe that in the case of this experiment they can be understood not only as statistical false positives but also as biological false positives but statistical true positives. Since each criterion is computed over a long period of recording, they can become “significant” due to processes such as drift of the sample or bleaching of the fluorophore but not due to a real change in activity. Thus, ROIs can drift away from the focal plane which can result in increase or decrease of signal. This interpretation is supported by the fact that criterion 2 was “significant” more often than criteria 1 and 3 (Fig. 6e) because the sample had more than twice as much time to drift during the period over which criterion 2 was computed. Importantly, these uncontrolled processes are equally likely across experimental group because animals were tested in a

random order, and the occurrence of false positive barcodes should not be, and in fact is not, different across groups. That being said, this is even more remarkable that the only cluster with different occurrence across groups was “0-0+” because it has the opposite signs for criteria 2 and 4, which is unlikely to result from drift.

In view of the above, we did not account for multiple comparisons because we did not set out to determine statistical significance of the criteria in a true sense. If we accounted for multiple comparisons, thresholds used to compute ternary barcodes would be dependent on number of detected ROIs in each fish. As a result, two ROIs with identical response profiles from two different fish might end up in different clusters simply because in one fish there were more ROIs than in the other. Nevertheless, we agree that it is very important to ensure that the key finding of this experiment (Fig. 6e_{ii}) is not sensitive to the threshold value used during the bootstrapping procedure. As shown in the figure below, we have tested a range of thresholds (from conservative 0.5 % to liberal 15 %) and found that, as expected, the number of ROIs assigned to clusters with non-zero criteria increases with the threshold because more liberal thresholds allow for more non-zero criteria. However, for all tested thresholds, the key finding remained qualitatively the same: the *only* cluster, whose occurrence was significantly different across groups, was “0-0+”.

Figure 1: The key finding of Fig. 6 is not sensitive to threshold for binarizing criteria into barcodes

a, Thresholds used to convert 4 criteria into ternary barcodes. Probability density functions represent a null-distribution of a criterion of one hypothetical ROI. To assign barcodes to ROIs using a threshold t , a criterion was replaced with “-” if it was less than $t/2$ th percentile of the null-distribution, with “+” if it was greater than $(100 - t/2)$ th percentile, and with “0” otherwise. This results in 81 possible clusters (3^4), most of which, however, do not contain any ROIs.

b, Within-fish fractions of ROIs labelled with different barcodes (similar to Fig. 6e_{ii} of the manuscript) defined using different thresholds. As in Fig. 6e_{ii}, only those clusters that contain, on average, at least 2 % of ROIs in at least one experimental group are shown. Magenta rectangles indicate “0-0+” cluster.

c, P-values of inter-group comparisons using Kruskal-Wallis test performed on each cluster. For all thresholds, the only cluster whose occurrence depends on experimental group was “0-0+” (for 10% threshold, the p-value is 0.0485).

In summary, we are hoping that, in view of the above, we have reached a compromise with Dr. Ehrlich's request to correct for multiple comparisons, by implementing the following changes to the manuscript with regard to this major concern:

- We have removed the term "statistical significance" from corresponding sections and explicitly clarified that bootstrapping was used only to define thresholds for converting criteria into ternary barcodes.
- We have mentioned in the Methods that the key finding was not sensitive to changes in threshold value used for computing barcodes.
- As requested, we have listed the number of "0-0+" ROIs as well as total number of ROIs in each group in the Results.

Minor concerns

Are the colors in Figure 2ei flipped? This panel conflicts with data in Figure 2ci and gain 0 in Fig 2ei doesn't match infinite lag in Fig 2eii.

The colors were indeed flipped and are now fixed. We thank the reviewer for alerting us on this!

The authors should speculate on which nodes in their control diagram might be under the influence of cerebellar/internal model output?

As requested by reviewer 2, we performed an additional experiment, in which we asked if one the possible actions of the internal model is to modify the time constants of optomotor integrators (as was proposed for the oculomotor system [Glasauer, 2003; Sanchez et al., 2018]). The results presented in revised Fig. 7 suggests that this option is plausible. Nevertheless, we discuss an alternative that the internal model might affect the parameters of the pre-motor circuits. Importantly, these options are not mutually-exclusive.

L215 - "metric of tiredness" explanation doesn't enhance understanding of the model. Relatedly, L443: "model suggests... duration is determined by the interplay between... sensory drive and tiredness." Anthropomorphizing needs more explanation if the authors feel it justified.

We believe that this anthropomorphizing can make our discussion of how the model can explain all subtleties of acute reactions more intuitive. For example, increased interbout duration under very low gain conditions (Fig. 2d,) can be intuitively explained by the following logic: very low gain -> very long bouts -> fish gets very "tired" -> it needs more time to "rest" before the next bout. After careful consideration, we have decided to keep this term but, accordingly, added more explanations.

A brief description of model parameter solutions would be informative. For instance, how do ω_f and ω_r compare? How do τ_v and τ_m compare?

We have added a panel to Extended Data Fig. 1 with requested information.

I would expect "velocity integrator" ROIs to exhibit a slower return to baseline after grating stimulus offset, but Figure 3g appears to show similar decays for "sensor" and "integrator" ROIs.

The authors should show activity aligned to stimulus offset, similar to Figures 3h and 3i. If integrators do not relax more gradually, how does one reconcile differing onset and offset kinetics?

We were defining integrators using the fluorescence only up to 4 s after the grating onset in each trial while ignoring the decay period after the offset. In the revised manuscript, we have changed the method and are now considering the whole fluorescence trace. As expected, sensory ROIs with longer time constants not only rise more slowly but also return back to baseline more slowly (Fig. 3 g-i). This change of definition had minimal effects on the spatial distribution of sensory ROIs and did not affect the conclusions (Fig. 3j).

Figure 6c - Non-adapting animals don't appear to exhibit an acute response. In this case, a more accurate description would be "non-responding" than "non-adapting." This is critical, because their response to the lagged reafference cannot be attenuated if it isn't existent in the first place. The

authors should show evidence, if existent, that the “non-adapting” group has an elevated bout duration during trials 21:30.

We have used the terms “adapting” and “non-adapting” because the metric used to define the groups was one of the metrics of long-term adaptation (reduction of acute reaction during the adaptation). Extended Data Fig. 5b_{ii} and c, shows that there is indeed very little (but nevertheless detectable) acute reaction in the “non-adapting” group. However, we think that the term “non-responding” can also be misleading, as it may imply that they did not show the optomotor *response*, which is not true. Perhaps the correct term would be “not reacting to changes in visual reafference” but this seems cumbersome. We hope to have reached a compromise by explicitly mentioning this concern in the text: “It is important to note that non-adapting lag-trained animals not only failed to exhibit long-term adaptation (Extended Data Fig. bii, cii-iii), but also showed a barely detectable acute reaction (Extended Data Fig. 5bii, ci). Therefore, the more correct label for this group would be “not reacting to changes in visual feedback”. We use the term “non-adapting” for convenience.”

The authors should indicate in Experimental protocols when the light sheet was illuminated with respect to behavior during imaging sessions.

As now indicated in the revised Methods section, “Larvae were continuously illuminated with the excitation blue light throughout the entire imaging sessions, which could potentially interfere with the behavior. However, behavior performed under the light-sheet microscope was overall comparable with the behavior performed in the behavioral rigs. Thus, for example, normal reafference control larvae performed, on average, 12.1 ± 0.4 bouts per trial in the behavioral rigs and 13.5 ± 3.0 under the light-sheet microscope (mean \pm SEM across larvae, data not shown), and the shape of the bouts was similar (compare Fig. 3c, Fig. 6di and Extended Data Fig. 6a with Fig. 1d and Fig. 4b). Furthermore, acute reaction and long-term adaptation effects were also present under the light-sheet and comparable with the behavioral rigs (compare Fig. 6c and Extended Data Fig. 5 with Fig. 4c-f).”

The control theoretic term “plant” deserves an introduction in line 41.

We have decided to remove this term from the manuscript and provided a clearer explanation.

More detail would be helpful regarding computation of bout power. How was the “tail trace” derived, and what are its units? (L1059).

Computation of bout power has been clarified, and the units of bout power (rad^2) have been added in the Methods and figures.

The lagged group purportedly became “statistically indistinguishable from the control group... (Fig 4e),” but that panel contrasts groups on the difference between adaptation and baseline levels, not on their durations directly.

We have updated Figure 4e and corresponding text according to the first major concern.

It is unclear how the data in Extended Figure 6 evidence the figure’s title. Are the arrows and color coding meant to indicate lack of correlation between the behavior and activity? If so, the relationship should be shown more explicitly.

This has been clarified. As now written in the revised Results section: “To test whether activity of the 0-0+ ROIs could be explained simply by motor activity of the larvae, we modelled an artificial ROI, which linearly encodes motor activity (motor regressor) and processed it in exactly the same way as we processed the real ROIs (Extended Data Fig. 6a). As expected, bout-triggered responses of the motor regressors were markedly similar to the behavior of the larvae (Extended Data Fig. 6b-c), showing an acute increase in response to unexpected presentation of lagged reafference (dark-blue arrows in Extended Data Fig. 6a-c). Importantly, this acute reaction was not observed in response profiles of 0-0+ ROIs (Extended Data Fig. 6d). Finally, we were able to find ROIs whose activity actually showed an acute reaction and was in general similar to the motor regressor (Extended Data Fig. 6e, see also activity profile of the ROI 2 shown in Fig. 6d in blue). Therefore, observed dynamics

of bout-triggered responses of 0-0+ ROIs cannot be explained by a motor component and rather reflect the output of a recalibrating internal model.”

-signed

David Ehrlich

Reviewer comments

We are very grateful to David Ehrlich and to an anonymous reviewer for their careful reading of this manuscript, for highlighting its importance and other strengths as well as for their insightful and valuable comments and suggestions. We have noted their contribution in the acknowledgements. The questions and requested revisions have been carefully considered and implemented to improve the manuscript.

Reviewer #2

General assessment and a summary of any substantive concerns

This manuscript by Markov and colleagues presents a study revealing how the zebrafish adjusts to visual feedback, thereby adjusting its own motor output in terms of frequency of swim bouts and their vigor. This study is a rigorous, well designed behavioral and imaging study that shows that zebrafish not only respond with predictable behavioral changes when this reafference is changed acutely but also that zebrafish learn over time to expect specific visual feedback. In summary, the authors make the claim that acute reaction to changed visual feedback can be implemented by a 'simple' feedback controller that integrates the optic flow, which they claim to reside in specific brain areas (e.g. pretectum, thalamus) as evidenced by light-sheet functional calcium imaging. However, if visual feedback is altered using a lag in visual feedback for a long period of time, fish learn to slowly adapt which the authors claim is proof of a learned internal model. Purkinje cells (PC) seem to play a role in this learning as their cell type-specific chemogenetic ablation impairs long-term adaptation, but not acute adaptation. Thus, using acute and long-term manipulations of the visual feedback, the authors demonstrate that the neural circuitry underlying the OMR is and can be calibrated and needs to be recalibrated when visual feedback is changed abruptly.

While the experiments of acute motor adaptations in head-embedded zebrafish are not new, the rigor and detailed investigation of behavioral consequences in response to subtly altering the visual reafference, provide new insights on how visual feedback works in this context and a great data resource for the specific field. E.g. the finding and distinction that the initial 220 ms of a swim bout is stereotyped ballistic period is followed by a reactive period in which fish can affect the tail beat amplitude is relevant for modeling purposes. The authors provide a convincing feed forward controller mechanism that may not require learning, according to their modelling and discussion. The cerebellum has long thought to integrate sensory and motor-related information in Purkinje cells (PC), the main computational components that disseminate information to the motor systems and enable motor learning. In this case, the authors show that PCs mediate the long term adaptation by generating a new transgenic line that allows ablation of (most?) PCs in a well-controlled set of experiments. While it would be important to determine the mechanism, this might be more suited for a future detailed study.

The research appears rigorous, with clear figures, legends, well-executed and presented data are extensive and appropriately analyzed with relevant statistical methods, and the materials and methods section is extensive, well written, detailed, and clear.

Despite the many strengths of the current manuscript, some questions remain and may require revisions or discussion in the paper:

- 1) Given their demonstrated high technical abilities, why did the authors not perform whole-brain light-sheet recordings using the protocol of long-term adaptation? (Under non-COVID-19 circumstances,) this would be an easy and interesting experiment to investigate brain-scale learning and adaptation in question, potentially transforming the findings of this study as it both might reveal non-PC cerebellar activity or distributed activity that might be part of the 'internal model'. At the very least please discuss.

We have performed the requested experiment but found only insignificant number of ROIs with properties of internal models and have included all this new data in the final Results

section. These ROIs were inconsistent across fish and lacked any apparent spatial organisation in both adapting and non-adapting groups. Therefore, we believe that the major role in encoding an internal model for long-term adaptation to modified visual feedback belongs to the cerebellum, which is in line with the result of our loss-of-function experiment.

Furthermore, we used these data to gain some insight on what could be the effects of a recalibrating internal model on the feedback controller involved in OMR. Specifically, we asked if one of the possible actions of the cerebellum is to modify the time constants of optomotor integrators (as was proposed for the oculomotor system [Glasauer, 2003; Sanchez et al., 2018]). The results presented in revised Fig. 7 suggests that this option is plausible. Nevertheless, we discuss an alternative that the cerebellum might affect the parameters of the pre-motor circuits. Importantly, these options are not mutually-exclusive.

- 2) The authors should provide supplemental movies of exemplary behavior and neural recordings under various different conditions to allow the reader to a) better understand the manipulations and b) provide deeper experimental insight of the findings.

We have provided two supplemental movies: one showing fish behavior under different refference conditions (gain, shunted and non-shunted lag, gain drop), and another one showing raw activity of two example ROIs (sensory and motor) in the brain during a whole-brain light-sheet imaging experiment.

- 3) The authors should include single-cell examples (in supplement if necessary but better in the main figure) of functional imaging- for critical classes of ROI and how they compare to single neuron recordings. The omission of these kinds of examples might otherwise be interpreted that ROIs cannot represent single cells or important details cannot be resolved.

We have added images of example ROIs shown in Fig. 3c-d, g-h (Fig. 3b). With our resolution of whole-brain imaging under the light-sheet microscope it is hard to clearly resolve single cell bodies in most of the brain regions. However, we believe that at least these four example ROIs correspond to cell bodies (Fig. 3b).

- 4) In Extended Data Fig. 1 a formal mathematical description of the controller model is provided, but it would be much preferred if the author would offer the associated Matlab code for all conditions and figures for reproducibility and rigor and future comparisons.

The data and code have been made publicly available and corresponding sections added to the manuscript.

- 5) It would be nice to see a verification of the PC ablation, Extended Data Figure 3 appears ambiguous whether some PC neurons, especially dorsally remain intact. The image does not seem to provide conclusive evidence that all of them are gone. However, the effect is strong. Therefore, please provide quantification and possibly additional verification using higher resolution images that all PCs are gone, at least in an example fish. Discuss in the text to aid clarity of the findings.

We have rephrased the corresponding section in the text, making it explicit that while it is possible that some of the PCs remain, their dendritic trees are completely destroyed. We have added a panel to the figure which hopefully illustrates this point better than the original panel. Finally, we have quantified the structural segregation of the PCs by computing local image entropy within the cerebellum as high entropy suggest complex anatomical structure of a healthy tissue whereas lower values indicate structural segregation and collapse. We chose this somewhat derived method of validating the ablation protocol (as opposed to, say, counting remaining cells) because the most prominent effect of the ablation was not the reduction in the number of fluorescent cell somata (although this effect was also clearly present) but rather segregation of the processes of the remaining cells into puncta, which is directly addressed by the local entropy metric.

- 6) Provided the strong evidence from behavioral data and neural recordings, the model in Figure 7 is relatively disappointing. Instead, no effort was made to integrate or quantitatively model the cerebellar/ PC role, compare and identify variables that account for the cerebellar internal model. Especially, PC specific model that would illustrate the predictions for perturbation experiments would be much preferred. For example, the authors could use the bout-triggered responses that gradually decrease during the adaptation period and increase back to the initial level when entering the post adaptation phase. A model that also would account for both proposed mechanisms should at least be attempted or discussed. One interesting point such a model could answer as well is that modeling realistic PC activation (even slow Calcium dynamics) might suggest a role for single PCs as well as for the population in terms of calibration/ learning.

We have tried to incorporate the results from the new experiments into the model and suggest explicit ways in which the cerebellar activity can alter the feedback controller mechanism.

- 7) In line 1162 and other places custom-written genetic algorithm. Please specify what that is. Like the point above, consider making the code available.
- 8) Please make all data or at least exemplary data available for better reproducibility.
- 9) Please make the associated code available for better reproducibility.

All data and code have been made publicly available and corresponding sections added to the manuscript.

- 10) Why is there no mechanistic discussion of overall PC circuit function?

We have expanded the discussion of PCs and how they may affect internal models.

- 11) Please expand discussion including relevant work in other species e.g. Kostadinov et al. 2019, Heffley, W. et al. 2018. But also, how these findings fit into recent work in zebrafish involving the cerebellum. Lin et al. 2020.

We have done this in the current version of the Discussion.

- 12) In the discussion, could the authors discuss how acute changes in afference blend with the internal model over time? The authors did not convincingly make the case that these two mechanisms are completely separate in intact animals, and it is expected according to their data that updating of the internal model begins after one bout.

We have attempted to explain this better in the current version of the Discussion.

Overall, the data appears to be of high quality, and the authors have gone to great lengths to establish extensive behavioral data sets for various incremental variations of visual reafference, provided large statistical samples of neurons, repetitions, and animals in each condition presenting a statistically sound study throughout. Especially, the behavioral data provides excellent evidence for its conclusions that different neural mechanisms are at play. To the best of my knowledge, both the behavioral results, modeling, transgenic fish lines for PC ablation and imaging are all completely novel and this study represents both an important data resource for the specific field (e.g. answering how zebrafish respond to variations of acute and long-term visual feedback). More generally, the paper represents an advance in understanding whether and how internal model-based learning might occur or and will likely influence future studies on this topic.

If the above points would be addressed and a couple of cosmetic changes listed below would be made, the presented research is a good fit for Nature Communications. Both the insights into the visually guided behavior, visual reafference, and implication of cerebellar function in adaptation and technical execution are excellent, justifying publication in this journal.

Minor comments

Line 41: 'body plant' should be changed to 'body plan'

Line 62: 'body plant' should be changed to 'body plan'

We have decided to remove this term from the manuscript and provided a clearer explanation.

REVIEWERS' COMMENTS

Reviewer #1 (Remarks to the Author):

The authors provided a highly responsive resubmission with clarifying analysis and discussion. My concerns have all been addressed.

-David Ehrlich

Reviewer #2 (Remarks to the Author):

This revised manuscript is much improved, clearer and all concerns were addressed or appropriately discussed and clarified. In their point-by-point rebuttal, the authors address and resolve all important concerns.

The state of the art work represents novel insights in how the vertebrate cerebellum contributes to calibrating neural circuits required for appropriate behavior generation.

In its revised form, this work contains important data for the field and is ready for publication.